# Assessment of simulations of a polar low with the Canadian Regional Climate Model

**Marta Moreno-Ibáñez**[1,2]*, **René Laprise**[1,2], **Philippe Gachon**[1,3]

**1** Centre for the Study and Simulation of Regional-Scale Climate (ESCER), University of Quebec in Montreal (UQAM), Montreal, Canada, **2** Department of Earth and Atmospheric Sciences, University of Quebec in Montreal (UQAM), Montreal, Canada, **3** Department of Geography, University of Quebec in Montreal (UQAM), Montreal, Canada

* marta.moreno-ibanez@outlook.com

**Data Availability Statement:** AVHRR channel 4 observations are available from EUMETSAT (https://navigator.eumetsat.int/product/EO:EUM:DAT:METOP:AVHRRL1), and MODIS channel 31 observations and VIIRS channel M15 observations

## Abstract

Polar lows (PLs), which are intense maritime polar mesoscale cyclones, are associated with severe weather conditions. Due to their small size and rapid development, PL forecasting remains a challenge. Convection-permitting models are adequate to forecast PLs since, compared to coarser models, they provide a better representation of convection as well as surface and near-surface processes. A PL that formed over the Norwegian Sea on 25 March 2019 was simulated using the convection-permitting Canadian Regional Climate Model version 6 (CRCM6/GEM4, using a grid mesh of 2.5 km) driven by the reanalysis ERA5. The objectives of this study were to quantify the impact of the initial conditions on the simulation of the PL, and to assess the skill of the CRCM6/GEM4 at reproducing the PL. The results show that the skill of the CRCM6/GEM4 at reproducing the PL strongly depends on the initial conditions. Although in all simulations the synoptic environment is favourable for PL development, with a strong low-level temperature gradient and an upper-level through, only the low-level atmospheric fields of three of the simulations lead to PL development through baroclinic instability. The two simulations that best captured the PL represent a PL deeper than the observed one, and they show higher temperature mean bias compared to the other simulations, indicating that the ocean surface fluxes may be too strong. In general, ERA5 has more skill than the simulations at reproducing the observed PL, but the CRCM6/GEM4 simulation with initialisation time closer to the genesis time of the PL reproduces quite well small scale features as low-level baroclinic instability during the PL development phase.

## 1. Introduction

The polar regions experience a variety of climate-related extreme events and high-impact weather conditions such as katabatic winds, blizzards, and polar lows (PLs) [1]. PLs are intense mesoscale maritime cyclones that develop between the poles and the main baroclinic zone, mainly during the cold season. Their diameter varies between 200 and 1,000 km, and their associated near-surface wind speed is over 15 m s[-1] [2]. PLs are short-lived phenomena, with

are available from NASA (https://ladsweb.modaps.
eosdis.nasa.gov/). The observations from surface
stations can been downloaded using MET Norway
Frost API (https://frost.met.no/index.html), and the
observation from drifting buoys can be requested
to Canada's ISDM centre (https://www.dfo-mpo.
gc.ca/science/data-donnees/drib-bder/index-eng.
html). The ERA5 global reanalysis from ECMWF is
available at https://www.ecmwf.int/en/forecasts/
datasets/reanalysis-datasets/era5. The Global Self-
consistent, Hierarchical, High-resolution
Geography Database (GSHHG), available at https://
www.ngdc.noaa.gov/mgg/shorelines/, has been
used to represent the coastlines. The divergent
colourmap used in Fig 2 is provided by the Texas
Advanced Computing Center at https://sciviscolor.
org/. The simulation output and the coordinates of
the manually obtained tracks are available at
Borealis, the Canadian Dataverse Repository (doi:
10.5683/SP3/6E3ITE).

**Funding:** This work was supported by the
Discovery Grant program of the Natural Sciences
and Engineering Research Council of Canada
(NSERC) under Grant 707337, by the project
"Marine Environmental Observation, Prediction and
Response" (MEOPAR) of the Networks of Centres
of Excellence (NCE) of Canada, by the UQAM's
Faculty of Sciences under the programme "faculty
financial support", and by the excellence
scholarship of the Trottier Family Foundation. The
operation of the supercomputer Beluga is funded
by the Canada Foundation for Innovation (CFI),
Ministère de l'Économie et de l'Innovation du
Québec (MEI) and les Fonds de recherche du
Québec (FRQ). The funders had no role in study
design, data collection and analysis, decision to
publish, or preparation of the manuscript. There
was no additional external funding received for this
study.

**Competing interests:** The authors have declared
that no competing interests exist.

lifetimes ranging from three to 36 hours [3]. They develop over the open water near the snow-covered landmasses or the sea-ice edge during marine cold air outbreaks (MCAOs). PLs are associated with severe weather conditions, including gale-force winds and heavy snowfall. These conditions can lead to large waves [e.g., 4], low visibility, snow avalanches, and icing on infrastructures. Therefore, PLs pose a threat to coastal populations, infrastructures, transport, and economic activities, and in some cases they have led to casualties [e.g., 5]. Producing accurate weather forecasts of PLs is thus critical to provide communities with enough time to prepare.

Weather forecasting in the polar regions remains a challenge since conventional observations are sparse, with weather stations being mainly concentrated along the coast [6], and data assimilation often fails to optimally use the available observational datasets [7]. The small temporal and spatial scales–horizontal and vertical scales of 100 km and 1 km, respectively –of PLs makes them particularly hard to forecast and to reanalyse [8]. Global reanalyses have low resolution (> 30 km of grid mesh), so they often fail to capture observed PLs. For instance, the reanalysis of the European Centre for Medium-Range Weather Forecasts (ECMWF) known as ERA-Interim [ERA-I, 9], which has a grid mesh of 0.75˚, fails to capture many PLs [10, 11]. The fifth-generation ECMWF reanalyses ERA5 [12], which has a grid mesh of 31 km and hourly outputs, captures more PLs than its predecessor [13]. Regional reanalyses such as the Arctic System Reanalysis [ASR, 14] are likely to be more adequate to represent PLs than global reanalyses given their higher resolution, and the fact that they are adapted to a particular region. For example, the first version of the ASR, which has a grid mesh of 30 km, captures more PLs than ERA-I [15]. Limited-area high-resolution atmospheric models are also a useful tool to study PLs since they can represent more PLs compared to the coarser reanalysis used as initial and boundary conditions [e.g., 10].

PL forecasting has been improved recently thanks to the development of high-resolution, non-hydrostatic atmospheric models. Compared to large-scale models, convection-permitting models (CPMs) provide a better representation of convection as well as surface and near-surface processes [16], which play an important role in the development of PLs. Indeed, Stoll et al. [17] found that, compared to the ECMWF global model HRES based on the Integrated Forecast System (IFS) cycle 32r3 [18], which has a grid mesh of 25 km, the regional model ARO-ME-Arctic [19], which has a grid mesh of 2.5 km, represented better the small-scale features associated with a PL such as individual convective clouds.

The emergence of high-resolution atmospheric models comes with its challenges. The increased resolution of the models requires that the model parameterisations be adapted to the resolution of the CPMs [16, 20]. In the polar regions, the parameterisation of surface fluxes needs to be optimised [21]. Furthermore, to make correct forecasts, atmospheric models need to be initialised with good observed conditions. Initial conditions uncertainties affecting the prediction of small-scale weather systems are mainly associated with convective and mesoscale instabilities [22]. The initial conditions of moisture at the mesoscale are especially significant for PL forecasting [23]. The initialisation time also seems to have an impact on the representation of PLs, as shown by case studies of the PL developed on 3 March 2008 [24, 25]. McInnes et al. [24] found that the simulations with the MetUM using a grid mesh of 4 km showed better performance when the simulations were initialised at around 42 hours before the PL formed compared to the simulations initialised 24 hours later. The authors argued that this could indicate that initialising the simulations at an earlier stage may be necessary to reproduce the synoptic-scale atmospheric conditions leading to the PL development. Nevertheless, Wagner et al. [25] obtained opposite results using the Polar Weather Research and Forecasting (WRF) model with a grid mesh of 2 km. In effect, the authors found that the simulations that performed better were those whose initialisation time was closer to the genesis time of the PL.

In this work we conducted a case study of a PL that developed over the Norwegian Sea on 25 March 2019 with two main objectives:

1. To quantify the impact of the initial conditions on the simulation of the PL;

2. To assess the skill of the developmental version of the convection-permitting Canadian Regional Climate Model version 6 (CRCM6/GEM4) at reproducing the observed PL.

The main verification method used in case studies of PLs is visual verification, but this type of verification does not quantify the skill of the model [8]. Therefore, we have applied an objective method to verify the simulations of the PL against conventional observations. Since the PL made landfall in Norway, we have been able to use near-surface observations of a wide range of atmospheric variables. Given that more work is needed on the verification of near-surface fields in the polar regions [7], this study will partly contribute to fill in this research gap.

The article is organised in four sections. Section 2 provides information about the CRCM6/GEM4 and the datasets used for the verification of the simulations, as well as a description of the methods used to prepare the datasets and to verify the simulation output. Section 3 provides a description of the life cycle of the PL and includes the analysis of the results. Section 4 summarizes the main conclusions of this study.

## 2. Data and methods

### 2.1 Datasets

**2.1.1 Simulations.** The PL that developed over the Norwegian Sea on 25 March 2019 has been simulated with the developmental version of the convection-permitting CRCM6/GEM4. The dynamical core of the CRCM6/GEM4 has been developed from the limited-area version of the Global Environmental Multiscale Model [GEM; 26–28]. The CRCM6/GEM4 uses the dynamical core of the version 4 of the GEM model (GEM4), whose detailed description is given by Girard et al. [29]. GEM uses an implicit semi-Lagrangian method for spatiotemporal integration [26, 29]. The model uses a rotated longitude-latitude grid in the horizontal [30]. The vertical coordinate is a hybrid log-hydrostatic pressure coordinate, based on the formulation of hydrostatic pressure developed by Laprise [31]. For the spatial discretization, the model uses three-dimensional staggered grids, the Arakawa C grid in the horizontal and the Charney-Phillips grid in the vertical. For the lateral driving, GEM employs the nesting technique suggested by Davies [32], which consists of applying a sponge zone around the domain with a relaxation coefficient decreasing from the outside to the inside.

For the simulations reported here, the model uses a grid spacing of 0.0225° ($\approx$ 2.5 km), a vertical grid with 62 levels, and a time step of one minute. The size of the domain is 1024 x 1024 grid points (Fig 1), including the ten grid point sponge zone around the perimeter of the domain, and the model top is at 2 hPa. The output of the simulations, excluding the sponge zone, therefore covers an area of approximately 2510 x 2510 km$^2$. Such domain is sufficient to capture not only the mesoscale phenomena, but also synoptic-scale features affecting polar low development.

The following subgrid parameterisation schemes have been used for the simulation: the correlated-k radiation scheme [34], the planetary boundary layer scheme MoisTKE that unifies turbulence and cloudiness [27, 35], the non-convective condensation scheme Predicted Particle Properties [P3; 36], and the land-surface scheme Interactions between Soil, Biosphere and Atmosphere [ISBA; 37]. Since convection is partially resolved, the deep convection scheme is turned off and only the shallow convection scheme Kuo-transient [27] is used. The orographic gravity wave drag and blocking, and non-orographic gravity wave drag schemes are also turned off.

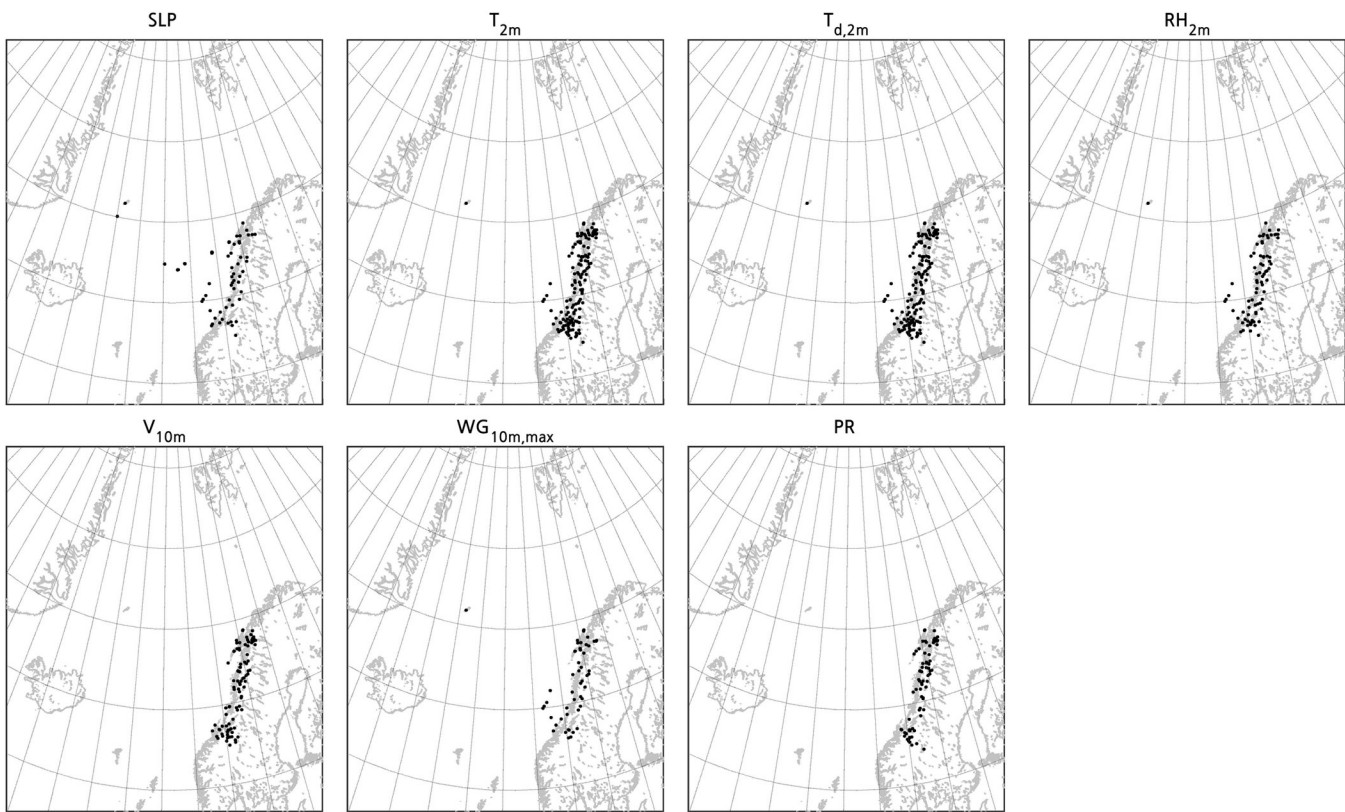

**Fig 1. Location of the drifting buoys and surface stations whose data has been used for the verification of the simulations of the PL.** The observations used are sea level pressure (SLP), 2-m temperature ($T_{2m}$), 2-m dewpoint temperature ($T_{d,2m}$), 2-m relative humidity ($RH_{2m}$), 10-m wind ($V_{10m}$), 10-m maximum wind gusts ($WG_{10m,max}$), and 1-h accumulated precipitation (PR). The region showed is the domain of the simulation excluding the sponge zone. The dataset used to plot the geographic contours has been obtained from the Global Self-consistent, Hierarchical, High-resolution Geography Database (GSHHG; [33]) under a CC BY license, with permission from Dr. Paul Wessel.

The atmospheric and ocean surface initial and boundary conditions have been obtained from the ERA5 global reanalysis, which has a horizontal grid of 0.25˚ [12]. From September 2007 onwards, ERA5 uses the Operational Sea Surface Temperature and Sea Ice Analysis (OSTIA) product for the sea surface temperature (SST), and the Ocean and Sea Ice Satellite Application Facilities (OSI SAF) product for the sea ice concentration (SIC). The CRCM6/GEM4 is hourly (daily) driven by the atmospheric (ocean surface) fields of ERA5. The ocean surface condition is temporally interpolated. The land surface initial conditions have been obtained from the Canadian Meteorological Centre analyses. Eight simulations were conducted by initialising the model every 6 hours from 23 March at 0000 UTC to 24 March at 1800 UTC. All simulations ended on 26 March at 0600 UTC. In what follows, we will refer to each simulation by its initialisation date; for instance, the simulation initialised on 24 March at 1200 UTC will be referred to as 24d12h.

Several variables at screen level have been output to compare them with conventional observations. It is important to note that the model computes the wind gusts using the wind gust estimate method developed by Brasseur [38]. This approach assumes that turbulent eddies lead to the downward deflection of air parcels located at higher levels in the boundary layer, producing surface wind gusts. Therefore, the mean wind and the turbulent structure of the atmosphere are included in the computation of wind gusts. This method provides a wind gust estimate as well as a bounding interval around this estimate. For this study, we use the instantaneous wind gust estimate that is output every hour.

**2.1.2 Conventional observations.** The simulations have been evaluated against hourly observations from weather stations provided by the Norwegian Meteorological Institute (MET Norway), and from drifting buoys provided by Canada's Integrated Science Data Management (ISDM) centre. Drifting buoys have been deployed by different international programs, the largest being the Global Drifter Program (GDP), which is the result of an international collaboration under the World Meteorological Organization (WMO) and the United Nations Educational, Scientific and Cultural Organization (UNESCO) umbrella. The GDP has been deploying surface Velocity Program Lagrangian drifters equipped with barometers that measure mean sea level pressure (SLP) every hour [39]. The main advantages of using conventional observations as "truth data" are that they directly measure meteorological variables and they have high temporal resolution, which is essential to capture PL development.

The observations from weather stations used to verify the simulations are SLP, 2-m temperature, 2-m dewpoint temperature, 2-m relative humidity, 10-m wind speed and direction, 10-m maximum wind gusts, and 1-h accumulated precipitation. The registered wind speed and direction are averaged over the last ten minutes before the observation time, and the maximum wind gust is the maximum wind registered during the ten minutes before the observation time. For drifting buoys, only SLP is available. Care should be taken when comparing the observed 10-m maximum wind gusts with the simulation and ERA5 wind gusts since the latter two are instantaneous wind gusts that are output every hour.

**2.1.3 ERA5.** The reanalysis ERA5 is produced by the EMCWF using a 4D-Var data assimilation scheme and the IFS Cy41r2 [12]. ERA5 has a grid spacing of 31 km and 137 levels to 0.01 hPa, and it provides hourly data. It covers the period from 1978 to the present, and there is a preliminary version from 1950 to 1978 [40]. Among other data, ERA5 assimilates conventional observations from surface stations and drifting buoys [see Fig 4 of 12]. Some studies have found that ERA5 shows a good performance in the Arctic [41, 42]. For example, Graham et al. [41] found that, compared to other reanalyses, including ERA-I, ERA5 had the smallest biases and root mean square errors (RMSEs), and the highest correlation coefficients at capturing the temperature, wind speed and specific humidity in the Fram Strait. Nevertheless, some studies have found limitations of ERA5 over Arctic sea ice. Since ERA5 does not represent a snow layer on top of the sea ice, the conductive heat flux from the ocean to the atmosphere is overestimated. As a result, like other reanalyses, ERA5 sea-ice surface temperature shows a warm bias during clear-sky conditions in winter [43]. This is consistent with the large warm bias of ERA5 2-m temperature over Arctic sea ice during the cold season compared to observations from drifting buoys [44].

## 2.2 Data preparation

We have prepared all the data from surface stations and drifting buoys available in the domain of the simulations in order to have complete time series of the variables whenever possible. Regarding surface stations, only data with acceptable quality has been selected, and outliers have been discarded. Therefore, some of the time series were incomplete either because there was already missing data or because some observations were discarded due to their low quality. In the case of noisy variables (10-m wind, 10-m wind gusts, and 1-h accumulated precipitation), the time series with one or more missing data have been completely discarded. In the case of smooth or continuous variables (SLP, 2-m temperature, 2-m relative humidity and 2-m dewpoint temperature), the time series with more than three missing values have been discarded. For the time series with three or less missing values, these values have been computed doing a linear temporal interpolation using the closest previous and following available observations, including sub-hourly observations. When the time between the previous or following observation and the missing observation was longer than one hour, the time series was discarded. Finally, since both wind speed and

direction are needed to verify the simulations, only the data of stations that provide both wind speed and direction have been retained for the verification of the wind field. In the case of drifting buoys, no time interpolation of the missing data has been done.

The simulated and ERA5 atmospheric fields have been spatially interpolated from the model grid to the observation points using either bilinear–for noisy variables–or bicubic interpolation–for smooth variables. A simple height correction has been applied to the simulated and the ERA5 temperature and dew point temperature to account for the difference in height between the real topography and the topography of the model. The lapse rate of the simulations and ERA5 at the lowest levels of the atmosphere has been used for the height correction of their respective temperature fields.

### 2.3 Verification

First, the track, size and intensity of the PL captured by the simulations and ERA5 have been compared to that of the observed PL. The track of the observed PL has been manually obtained using IR radiance satellite images from the Moderate Resolution Imaging Spectroradiometer (MODIS), the Advanced Very High Resolution Radiometer (AVHRR/3) and the Visible Infrared Imaging Radiometer Suite (VIIRS). The coordinates of the centre of the observed PL have been estimated at each hour from the genesis until the dissipation of the PL. The track that has been initially obtained using the satellite images has been improved by ensuring that the track is consistent with the conventional observations of SLP and 10-m wind. The tracks of the PL in the simulations and ERA5 have been manually obtained using the SLP field. The criteria to identify the beginning of the PL is the presence of at least three SLP closed contours in a map showing the SLP isobars every 1 hPa. The size of the PL has been estimated in all the datasets by measuring the diameter of the cloud signature during the mature stage of the PL. The intensity of the PL in the simulations and ERA5 is given by the SLP minimum at its centre. In the case of observations, the SLP minimum corresponds to the SLP observation from the surface station that is the closest to the centre of the PL. Since some stations are too far from the centre of the PL, only the observations from stations within a distance of 25 km from its centre have been considered.

Second, all the simulations and ERA5 have been verified against observations from surface stations and drifting buoys affected by the PL. Therefore, we have only used observations obtained within a distance of 300 km– which approximately corresponds to the radius of the cloud signature of the PL at its mature stage –from the centre of the observed PL. The total number of observations used are 352 for SLP, 860 for 2-m temperature, 820 for 2-m dewpoint temperature, 483 for 2-m relative humidity, 534 for 10-m wind, 318 for 10-m wind gusts, and 448 for 1-h accumulated precipitation. From the genesis of the PL until 25 March at 11:00 UTC or 12:00 UTC, depending on the variable, the number of observations is no more than 10, or there are no observations at all. The number of observations notably increases when the PL gets closer to the Norwegian coast (Fig 1). Therefore, the results are mainly representative of the mature and dissipation stages of the PL. The statistics computed to measure the performance of the simulations are the mean error (ME), the root mean square error (RMSE), and the Spearman correlation coefficient ($r$) [45]. Since the wind is a vector, the root mean square wind-vector-difference error has been computed [RMSE-WVD; e.g., 46]. The correlation coefficient has only been computed when at least three observations were available.

## 3. Results and discussion

### 3.1 Description of the life cycle of the PL

Northerly winds on the cold side of a synoptic-scale low located over the Barents Sea caused a MCAO in Fram Strait at the end of March 2019 (Fig 2A). Cold northerly winds to the west of

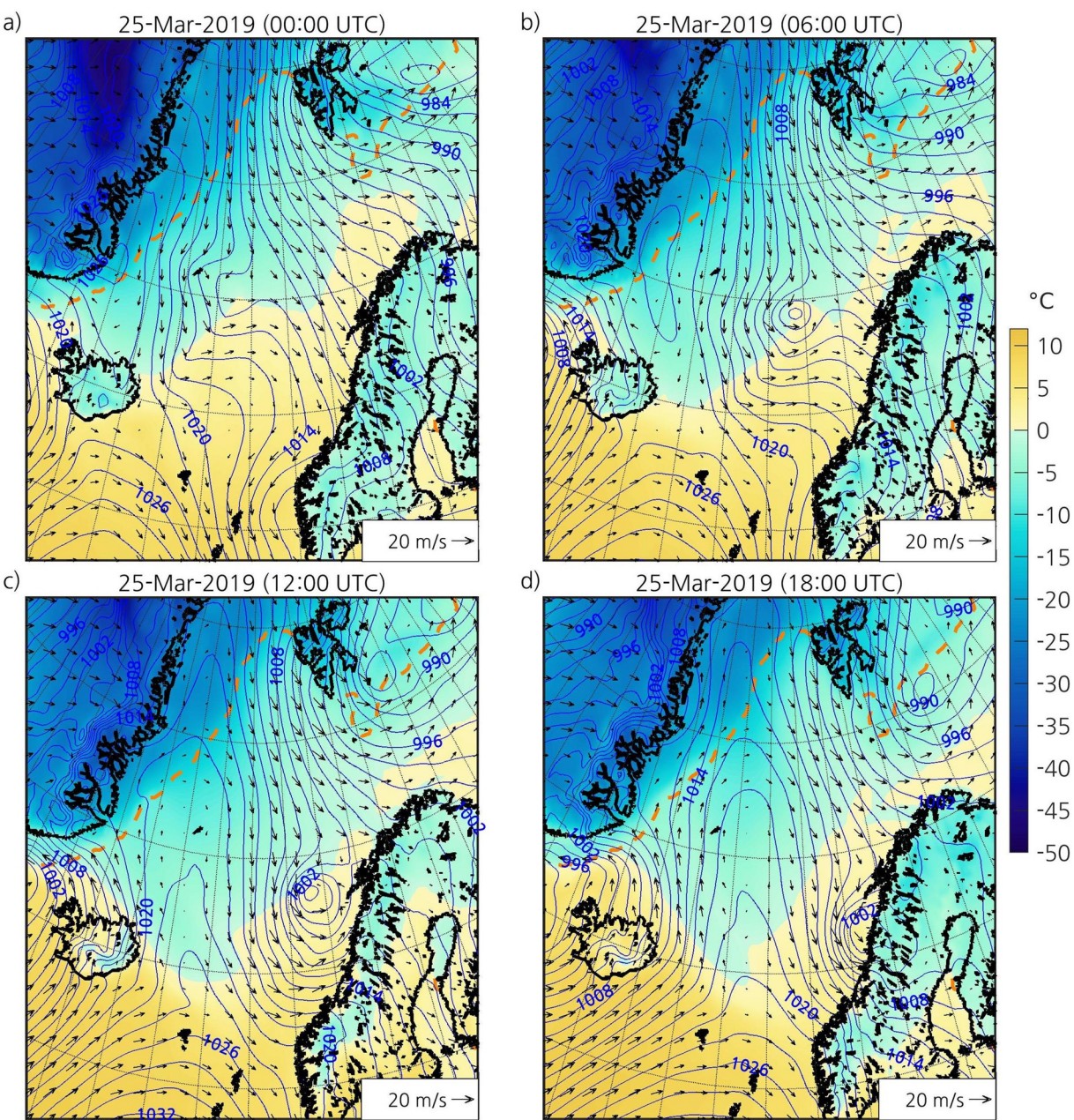

**Fig 2.** ERA5 atmospheric fields showing the development of the PL on 25 March 2019 at (a) 0000 UTC, (b) 0600 UTC, (c) 1200 UTC, and (d) 1800 UTC. The blue isolines represent the SLP (contours every 2 hPa), the colourmap represents the 2-m temperature (˚C), and the arrows represent the 10-m wind direction and speed, with longer arrows representing higher wind speeds. The orange dashed line represents the sea ice edge, which is defined as the 0.15 contour of the SIC corresponding to 25 March 2019 at 1200 UTC. The black outlining represents the coastline. ERA5 fields have been interpolated to the grid of the simulation using bicubic interpolation for the SLP and temperature and bilinear interpolation for the wind. The dataset used to plot the geographic contours has been obtained from the GSHHG [33] under a CC BY license, with permission from Dr. Paul Wessel.

synoptic-scale lows is a common favourable environment for PL development in the Nordic Seas [47]. The PL developed early on 25 March near the sea ice edge east of Greenland, in a region with a strong temperature gradient (Fig 2B). The cloud streets and open cells associated with the MCAO are visible on IR satellite images from the AVHRR channel 4 (Fig 3). The PL

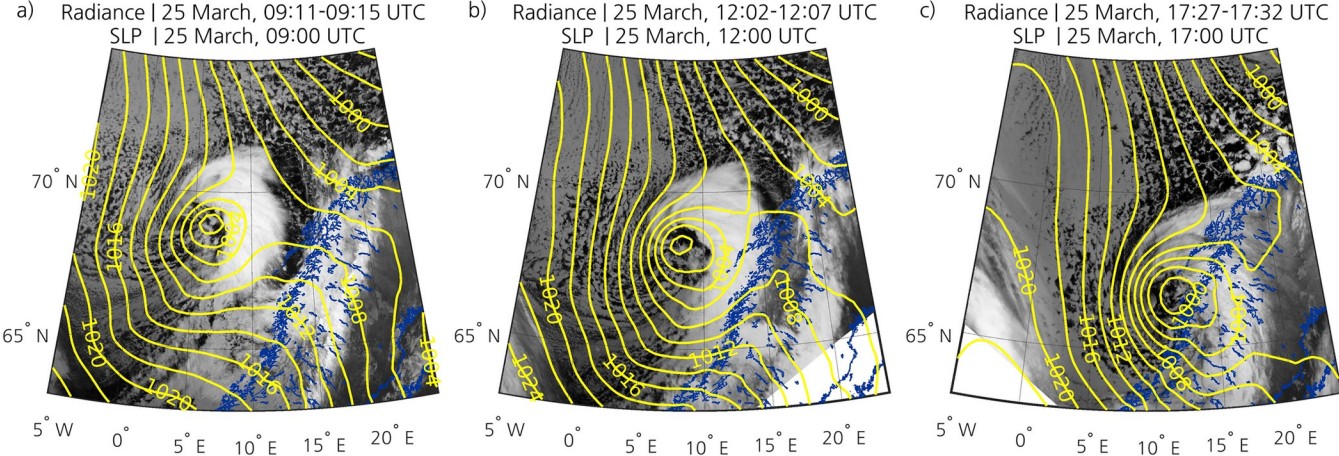

**Fig 3.** AVHRR channel 4 images showing the PL on 25 March (a) before it hits land, (b) when it is over a great part of the Norwegian coast, and (c) when it starts to dissipate. The yellow isolines represent the ERA5 SLP field (contours every 2 hPa). The blue outlining represents the coastline. The dataset used to plot the geographic contours has been obtained from the GSHHG [33] under a CC BY license, with permission from Dr. Paul Wessel.

started to form over open water at the leading edge of this MCAO, and a comma-shaped cloud signature was clearly identifiable in IR images by 25 March at 0200 UTC (not shown). Like many PLs in the Nordic Seas [e.g., 48], it moved southeastward as it deepened (Fig 2B and 2C). The PL hit land in Nordland county of Norway after 0900 UTC (Fig 3A). By 1200 UTC, it had reached a large part of the Norwegian coast (Figs 2C and 3B). The winds associated with the PL reached an observed maximum speed of 24.8 m s$^{-1}$. The PL started to dissipate at around 1800 UTC (Figs 2D and 3C). The lifetime of this PL was 20 hours (Table 1), in agreement with climatologies of PLs in the Nordic Seas [e.g., 15]. The estimated size of the PL at its mature stage was 586 km in diameter, which is larger than the typical diameter of PLs [e.g., 48]. With an average speed of 15 m s$^{-1}$, the PL travelled 1,070 km. Both the average speed and distance travelled are larger than those of most PLs [48].

## 3.2 Verification of the track, size and intensity of the simulated PL

Fig 4 shows the SLP isobars of the eight simulations and ERA5 on 25 March 2019 at 1500 UTC, when the mature PL was affecting the Norwegian coast. The large-scale features of the SLP field are similar for all simulations, whereas the simulations notably differ from each other at the mesoscale, in particular near the location of the PL. Overall, these spaghetti plots show that most simulations fail to represent the PL. Only 23d12h and the latest initialised simulations 24d12h and 24d18h represent a PL.

The PL in the the simulations and ERA5 forms and dissipates on 25 March. The simulations 24d12h and 24d18h represent well the size and lifetime of the observed PL, as well as the the

**Table 1. Lifetime, translation speed and distance travelled by the PL.**

| Dataset | Start hour [UTC] | End hour [UTC] | Size [km] | Average speed [m s$^{-1}$] | Distance [km] |
|---------|------------------|----------------|-----------|---------------------------|---------------|
| Observations | 0100 | 2100 | 586 | 15 | 1,070 |
| 23d12h | 1200 | 2000 | 402 | 14 | 395 |
| 24d12h | 0000 | 2000 | 585 | 12 | 892 |
| 24d18h | 0000 | 2100 | 561 | 15 | 1,113 |
| ERA5 | 0500 | 1900 | 561 | 13 | 636 |

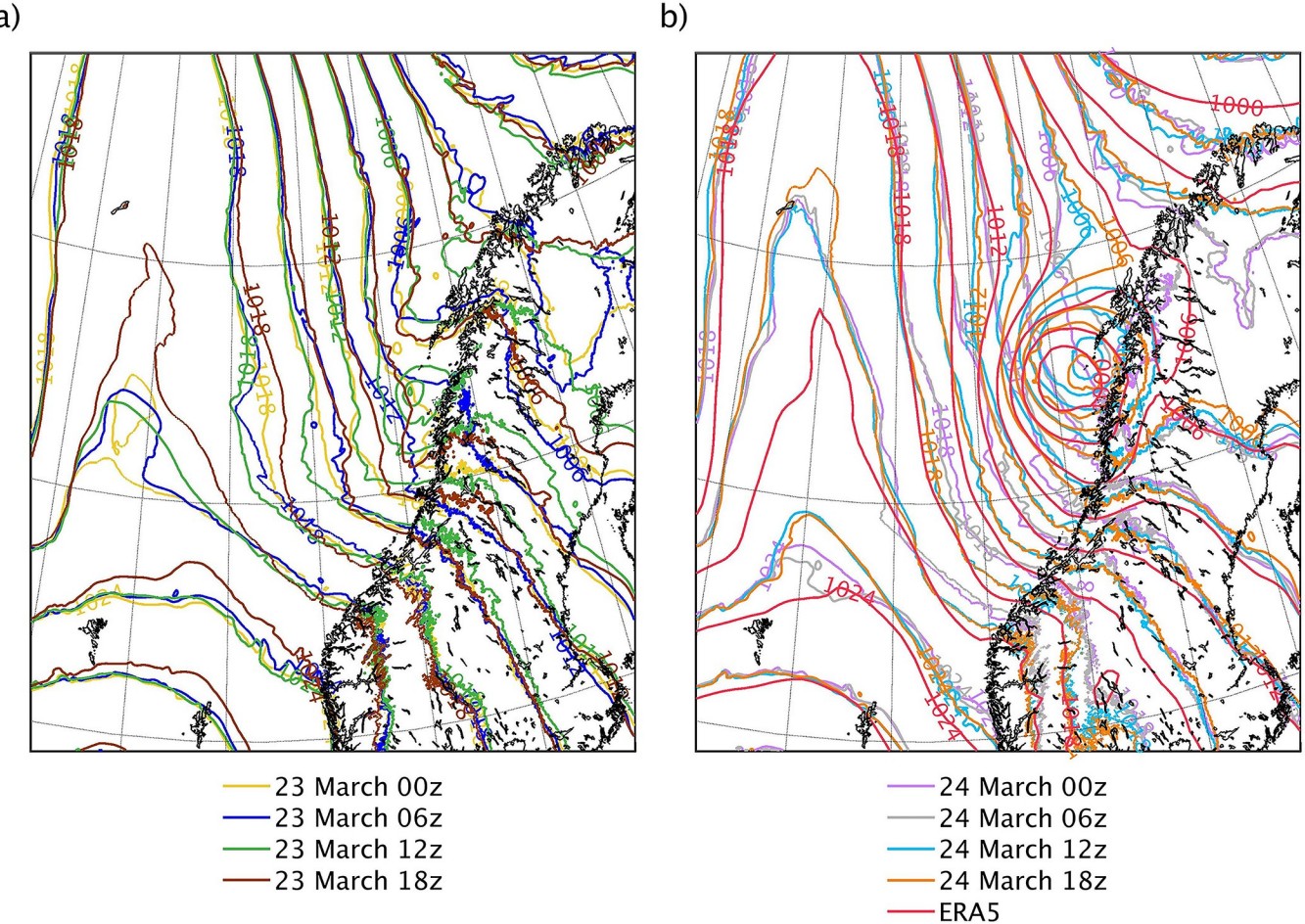

a)

b)

23 March 00z
23 March 06z
23 March 12z
23 March 18z

24 March 00z
24 March 06z
24 March 12z
24 March 18z
ERA5

**Fig 4. Spaghetti plots showing the SLP isobars (contours every 3 hPa) of the simulations and ERA5 on 25 March 2019 at 1500 UTC, when the mature PL was affecting the Norwegian coast.** The contour lines correspond to the SLP field of (a) the simulations initialised on 23 March at 0000, 0600, 1200 and 1800 UTC, and (b) the simulations initialised on 24 March at 0000, 0600, 1200 and 1800 UTC, and ERA5. The black outlining represents the coastline. The dataset used to plot the geographic contours has been obtained from the GSHHG [33] under a CC BY license, with permission from Dr. Paul Wessel.

timing of its genesis and dissipation (Table 1). The average speed of the PL in 24d18h is the same as that of the observed one, and the total distance travelled is similar. The average speed of the PL and the distance travelled are lower in 24d12h, but they are fairly close to the observed ones. The PL represented in 23d12h is much smaller than the observed PL, and its lifetime is less than half the lifetime of the observed PL. However, its average speed is similar to the observed one. The PL in this simulation forms eleven hours latter than the observed one, but it dissipates at a similar time. Therefore, the distance travelled by the PL in this simulation is significantly lower than the observed one. The PL in ERA5 has shorter lifetime and somewhat lower average speed than the observed one, but similar size.

The tracks of the PL in the simulations and in ERA5 are reasonably close to that of the observed PL (Figs 5 and 6A). The distance between the simulated and the observed tracks notably increases at the end of the lifetime of the PL, which may be partly due to the high uncertainty when determining the centre of the PL at its dissipation stage. The tracks in 24d12h and 24d18h remain within 100 km from the observed track most of the time, whereas the track in ERA5 remains within 50 km from the observed one. The observed SLP minimum attained near the centre of the PL is 999.1 hPa at 1500 UTC (Fig 6B). However, since the

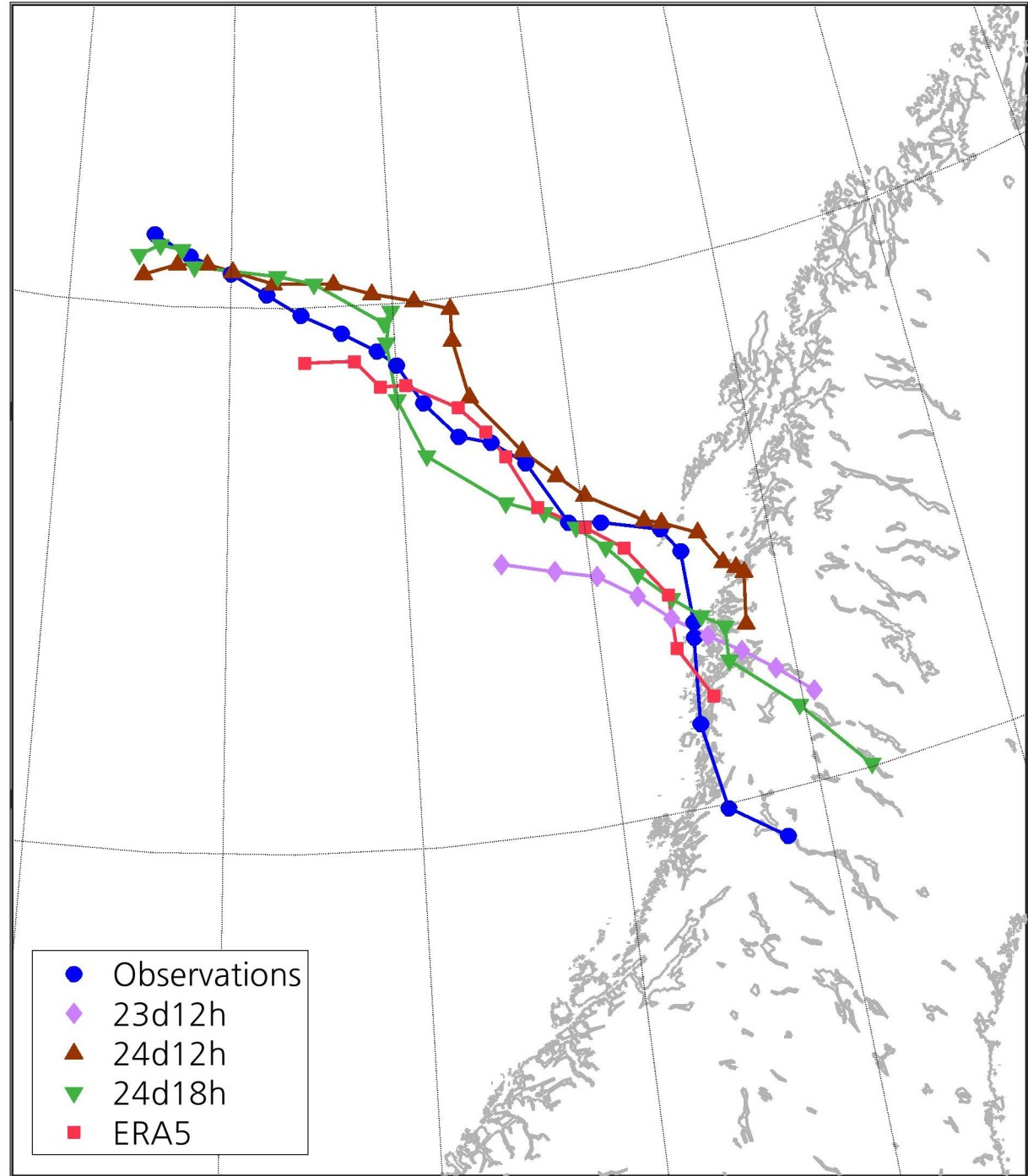

**Fig 5. Track of the observed PL and of the PL in the simulations and ERA5.** The markers represent the position of the PL at each hour. Information about the genesis and dissipation times of these PLs can be found in Table 1. The dataset used to plot the geographic contours has been obtained from the GSHHG [33] under a CC BY license, with permission from Dr. Paul Wessel.

surface station providing this observation is located 8.82 km from the centre, the real SLP minimum may be lower. The SLP minimum of the PL in 24d12h and 24d18h is 995.6 and 995.7 hPa, respectively, also at 1500 UTC. The PL in 23d12h shows a steeper decrease in SLP and, as

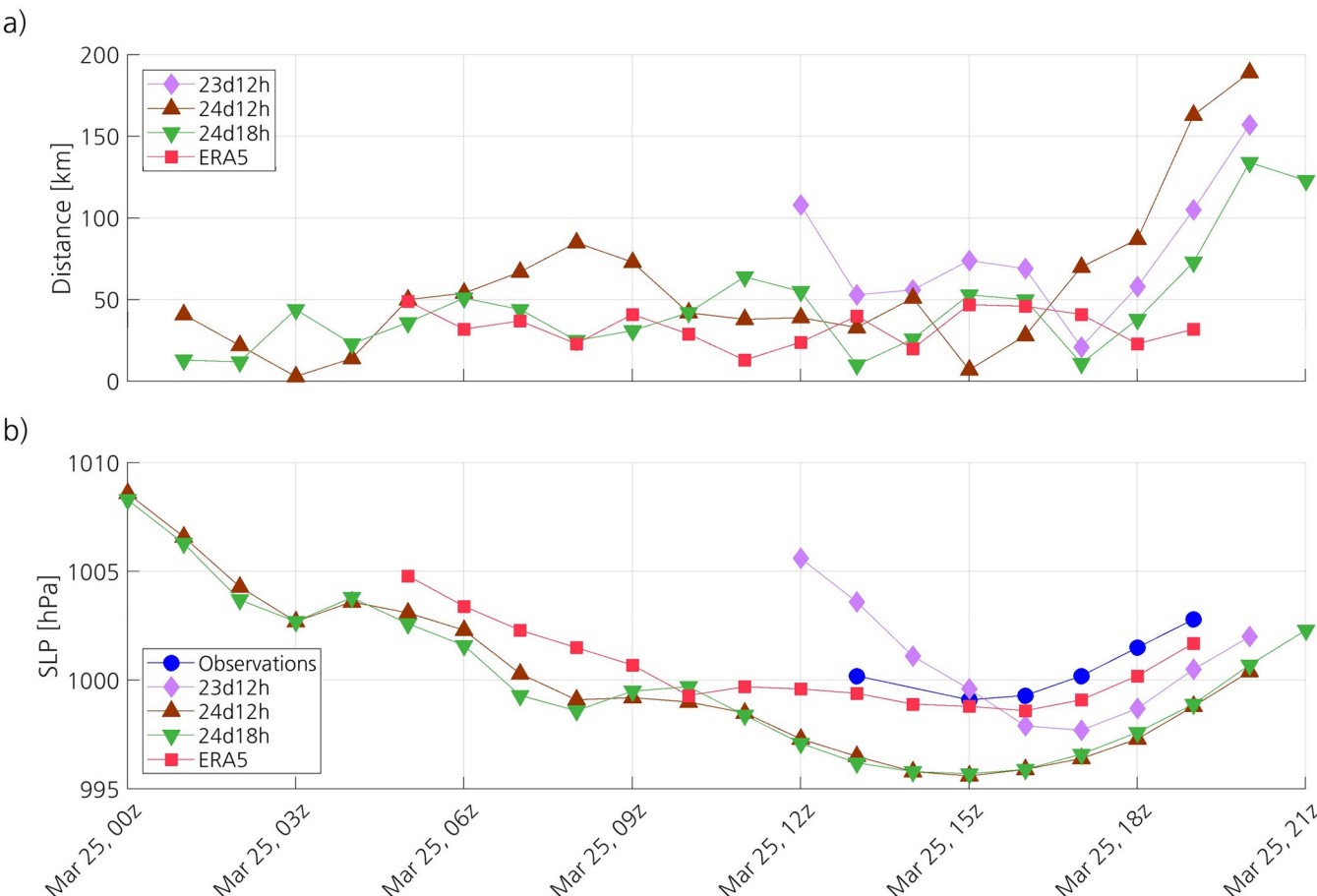

**Fig 6.** Time series of (a) the distance between the centre of the observed PL and the centre of the PL in the simulations and ERA5, and (b) the SLP at the centre of the PL in the simulations and ERA5, and the SLP observed at the surface station the closest to the centre of the observed PL.

a result, the SLP minimum is reached just two hours later than in the other simulations. With a shorter lifetime than the other simulated PLs, the PL in 23d12h deepens slightly less than the others and, therefore, its associated SLP minimum is closer to the observed one. Compared to the simulations, the PL in ERA5 deepens less, corresponding better to the observations. The evolution of the SLP at the centre of the PL in ERA5 follows closely the observations, and the SLP in ERA5 attains a minimum of 998.6 hPa at 1600 UTC.

### 3.3 Verification of the simulated PL against observations affected by the PL

**3.3.1 Sea level pressure.** As expected, all simulations except for 24d12h and 24d18h notably overestimate SLP, particularly the lowest observed SLP values (Fig 7). In contrast, 24d12h and 24d18h underestimate many SLP values between 1000 and 1010 hPa. The aggregate statistics show that 24d12h and 24d18h have lower absolute mean bias, higher accuracy and higher correlation coefficient than the other simulations (Table 2). Whereas both 24d12h and 24d18h have a ME of -0.2 hPa, the ME of the other simulations ranges from 2.1 to 3.5 hPa. The RMSE of 24d12h and 24d18h (2 hPa and 1.4 hPa, respectively) is considerably lower than that of the other simulations (between 3.2 hPa and 4.5 hPa). The Spearman correlation coefficient of 24d12h and 24d18h shows that they have, respectively, a strong and a very strong positive correlation with the observations. Except for 23d12h and 23d18h, which show a quite strong correlation with the observations, the other simulations show either weak or modest correlation.

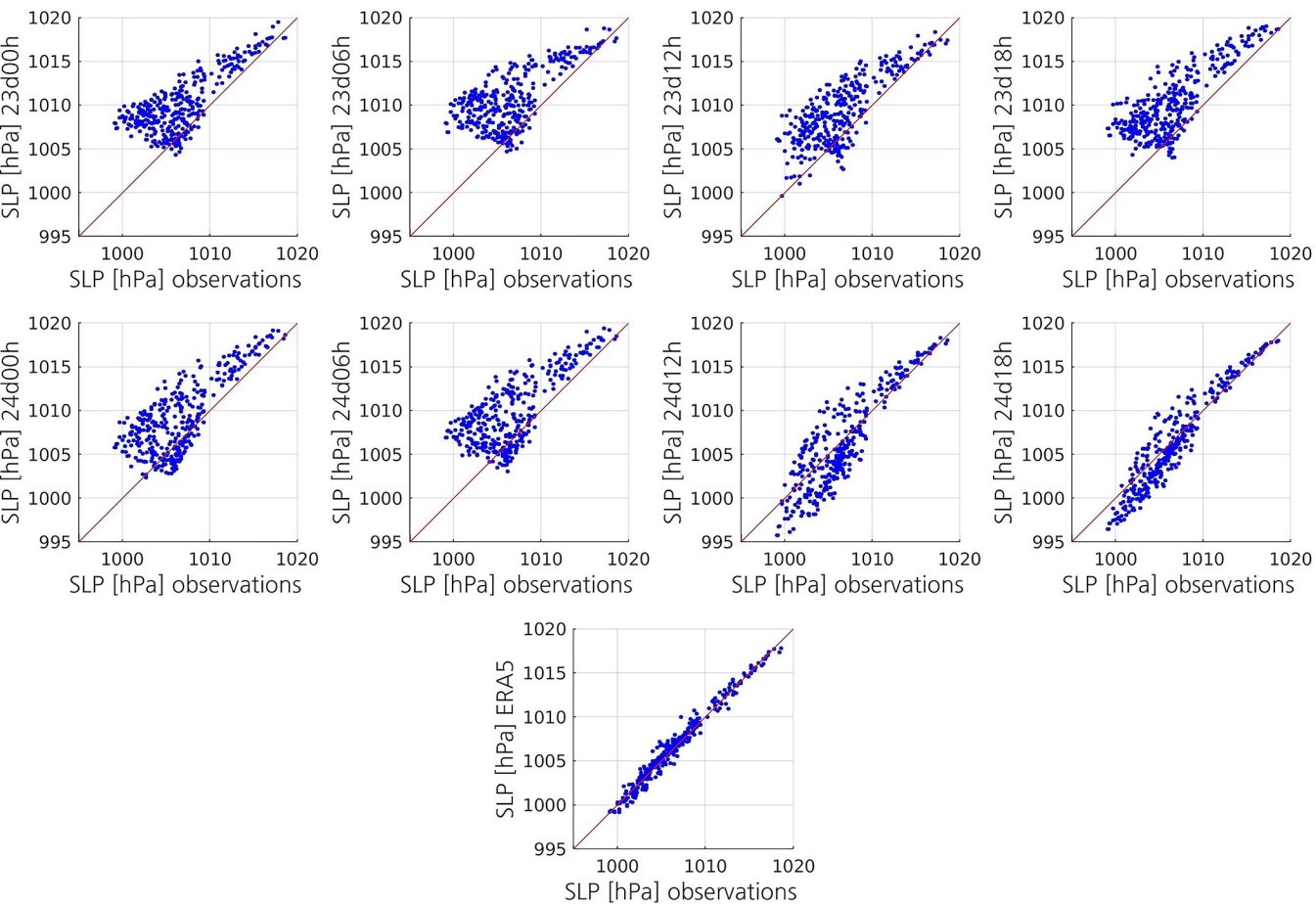

**Fig 7. Scatterplots displaying, for each simulation, the relationship between the simulated SLP and the SLP measured at surface stations and drifting buoys.** The scatterplot on the bottom displays the relationship between the ERA5 SLP and the observed SLP. The red line represents the regression line that would correspond to a perfect match between the values.

The simulations 24d12h and 24d18h have a small negative ME during the mature and dissipation stages of the PL, whereas the other simulations show a positive ME during the whole lifetime of the PL (Fig 8A). The ME of the latter steadily increases from around 1100 UTC until around 1600 UTC, which is likely due not only to the deepening of the PL, but also to the fact that its centre is getting closer to the surface stations. The time series of the RMSE of these simulations shows a similar pattern to that of the ME (Fig 8B). The main difference is that 24d12h and 24d18h also show an increase in the RMSE during the deepening of the PL, although it remains much lower than that of the other simulations. This decrease in accuracy, together with the negative mean bias, confirms the previous finding that the simulated PL in both simulations is deeper than the observed one. All simulations except for 24d12h and 24d18h show a significant decrease in the Spearman correlation coefficient from 1300 UTC until 1500 UTC, when it reaches a minimum, which corresponds to the time when the SLP minimum is observed (Fig 8C). Then, the correlation coefficient increases until the PL has dissipated. The simulation 24d12h shows a similar pattern, but the decrease in the correlation coefficient is much less pronounced. In contrast, 24d18h shows a small decrease in the correlation coefficient.

ERA5 shows better skill at representing SLP than the simulations, even than 24d18h. Although the absolute ME of ERA5 and 24d18h are both small (Table 2, Fig 8A), the

**Table 2. Aggregate statistics computed to verify the simulated and ERA5 SLP against the observations from surface stations and drifting buoys.**

|        | ME [hPa] | RMSE [hPa] | r |
|--------|----------|------------|------|
| 23d00h | 3        | 3.8        | 0.54 |
| 23d06h | 3.5      | 4.5        | 0.48 |
| 23d12h | 2.8      | 3.6        | 0.71 |
| 23d18h | 3.2      | 3.9        | 0.67 |
| 24d00h | 2.1      | 3.2        | 0.59 |
| 24d06h | 2.5      | 3.6        | 0.53 |
| 24d12h | -0.2     | 2          | 0.81 |
| 24d18h | -0.2     | 1.4        | 0.92 |
| ERA5   | 0.1      | 0.6        | 0.98 |

scatterplot shows that ERA5 has better skill (Fig 7). The higher accuracy of ERA5 is confirmed by its aggregate RMSE (0.6 hPa), which is less than half of that of 24d18h (Table 2). The RMSE of ERA5 remains relatively constant and less than one during the whole lifetime of the PL (Fig 8B). Like 24d18h, ERA5 shows a very strong positive correlation with the observations, but its correlation coefficient is slightly higher than that of 24d18h (Table 2, Fig 8C).

**3.3.2 Temperature at 2 m.** All the datasets have a positive temperature bias (Fig 9). The aggregate statistics indicate that 24d12h and 24d18h have higher mean bias and lower accuracy, but higher correlation coefficient, than the other simulations (Table 3). These two simulations have a higher ME (2°C) and a slightly higher RMSE (2.4°C) compared to the simulations that did not simulate the PL (ME between 0.8 and 1.3°C, and RMSE between 1.9 and 2.1°C). The simulation that captured a small and short-lived low, 23d12h, has a ME (1.7°C) and a RMSE (2.3°C) lower than those of 24d12h and 24d18h, but higher than those of the other simulations. All simulations show a quite strong correlation with the observations. Although 24d12h and 24d18h show the highest Spearman correlation coefficient (0.78), the difference between the simulations is very small.

In the initial stage of the PL, all simulations show a high positive ME, with 24d12h and 24d18h showing the highest bias (Fig 10A). This could indicate that the MCAO is not well simulated. However, there are only two observations available at 0100 and one at 0200 UTC, so the results should be interpreted with care. From 1000 UTC until the dissipation of the PL, the ME of 24d12h and 24d18h remains notably higher than that of the simulations that have not captured the PL. The ME of 23d12h remains lower than that of 24d12h and 24d18h, but higher than that of the other simulations virtually all the time. The important surface heat transfer from the ocean to the atmosphere that takes place in the simulations that have captured the PL (not shown) likely explains why these simulations have higher ME. The RMSE of 23d12h, 24d12h and 24d18h remains higher than that of the other simulations most of the time (not shown). The correlation coefficients of all simulations notably increase from around 1100 UTC until around 1500 UTC, and then the simulations show a strong or a quite strong correlation with the observations (not shown).

Surprisingly, the aggregate ME of ERA5 (0.9°C) is closer to that of the simulations that have not captured the PL than to that of the simulations that have captured it (Table 3). The ME of ERA5 remains significantly lower than that of 24d12h and 24d18h, and closer to that of the simulations that have not captured the PL, during the whole observed lifetime of the PL (Fig 10A). With an aggregate RMSE of 1.5°C, ERA5 has slightly higher accuracy than all the simulations (Table 3). The RMSE of ERA5 remains lower than that of the simulations during virtually the whole observed lifetime of the PL, and the difference in accuracy between ERA5

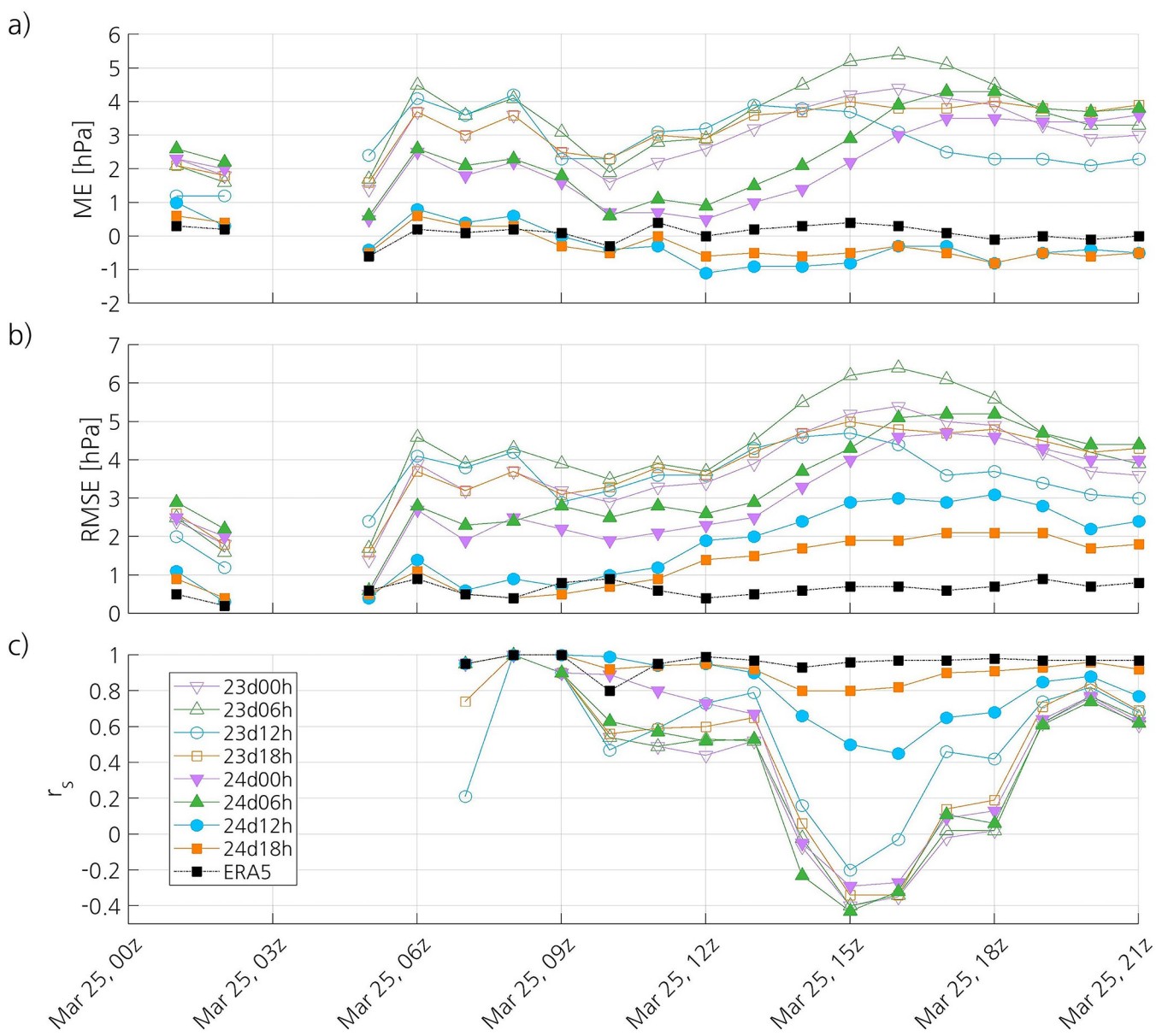

**Fig 8.** Time series of the (a) ME, (b) RMSE, and (c) Spearman correlation coefficient of the simulated and ERA5 SLP. The dataset used as reference to compute the ME is the SLP measured at surface stations and drifting buoys.

and the simulations is more important during its dissipation stage (not shown). ERA5 shows a strong correlation with the observations (0.87), which is higher than that of the simulations (Table 3). The time series of the correlation coefficient of ERA5 follows a pattern similar to that of the simulations, but it becomes higher than the correlation coefficients of the simulations during its dissipation stage (not shown).

**3.3.3 Dew point temperature at 2 m.** The aggregate statistics indicate that the simulations that have captured the PL do not show better skill at representing the dew point temperature than those that have not captured it (Table 4). Most simulations have lower ME than 24d12h and 24d18h (2.2˚C and 2˚C, respectively). All simulations have similar RMSE, ranging from 2.9 to 3.4˚C. Some simulations that have not captured the PL have weak correlation with the

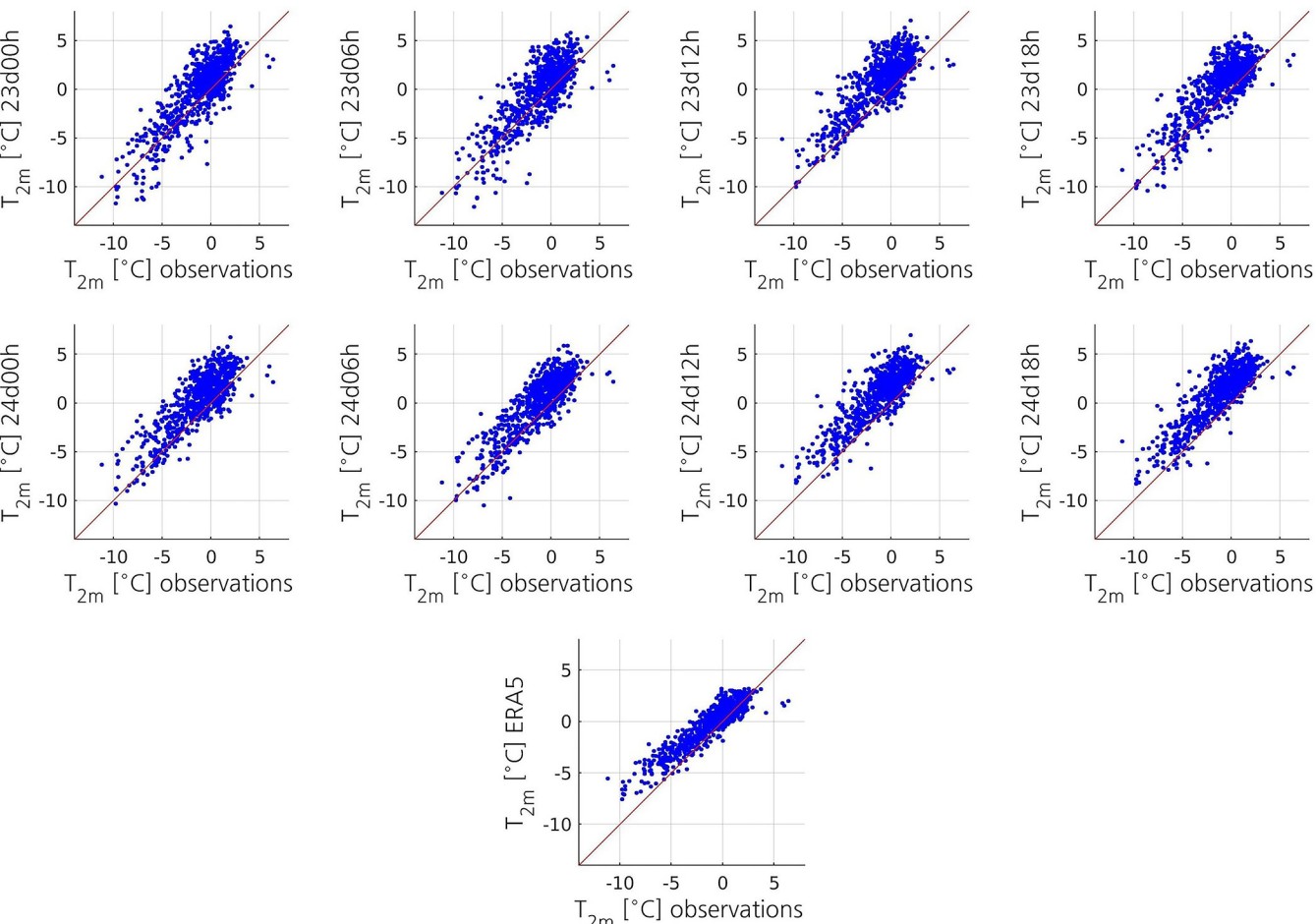

**Fig 9. Scatterplots displaying, for each simulation, the relationship between the simulated 2-m temperature and the 2-m temperature measured at the surface stations.** The scatterplot on the bottom displays the relationship between the ERA5 2-m temperature and the observed 2-m temperature. The red line represents the regression line that would correspond to a perfect match between the values.

observations, whereas the other simulations, including those that represent the PL, have modest correlation with the observations. Although the time series of the ME of each simulation is quite different, all the simulations show a positive ME during all or almost all the time

**Table 3. Aggregate statistics computed to verify the simulated and ERA5 2-m temperature against the observations from surface stations.**

|        | ME [˚C] | RMSE [˚C] | r |
|--------|---------|-----------|------|
| 23d00h | 1       | 2         | 0.75 |
| 23d06h | 0.8     | 1.9       | 0.74 |
| 23d12h | 1.7     | 2.3       | 0.74 |
| 23d18h | 1.2     | 2         | 0.73 |
| 24d00h | 1.3     | 2.1       | 0.75 |
| 24d06h | 1.2     | 1.9       | 0.77 |
| 24d12h | 2       | 2.4       | 0.78 |
| 24d18h | 2       | 2.4       | 0.78 |
| ERA5   | 0.9     | 1.5       | 0.87 |

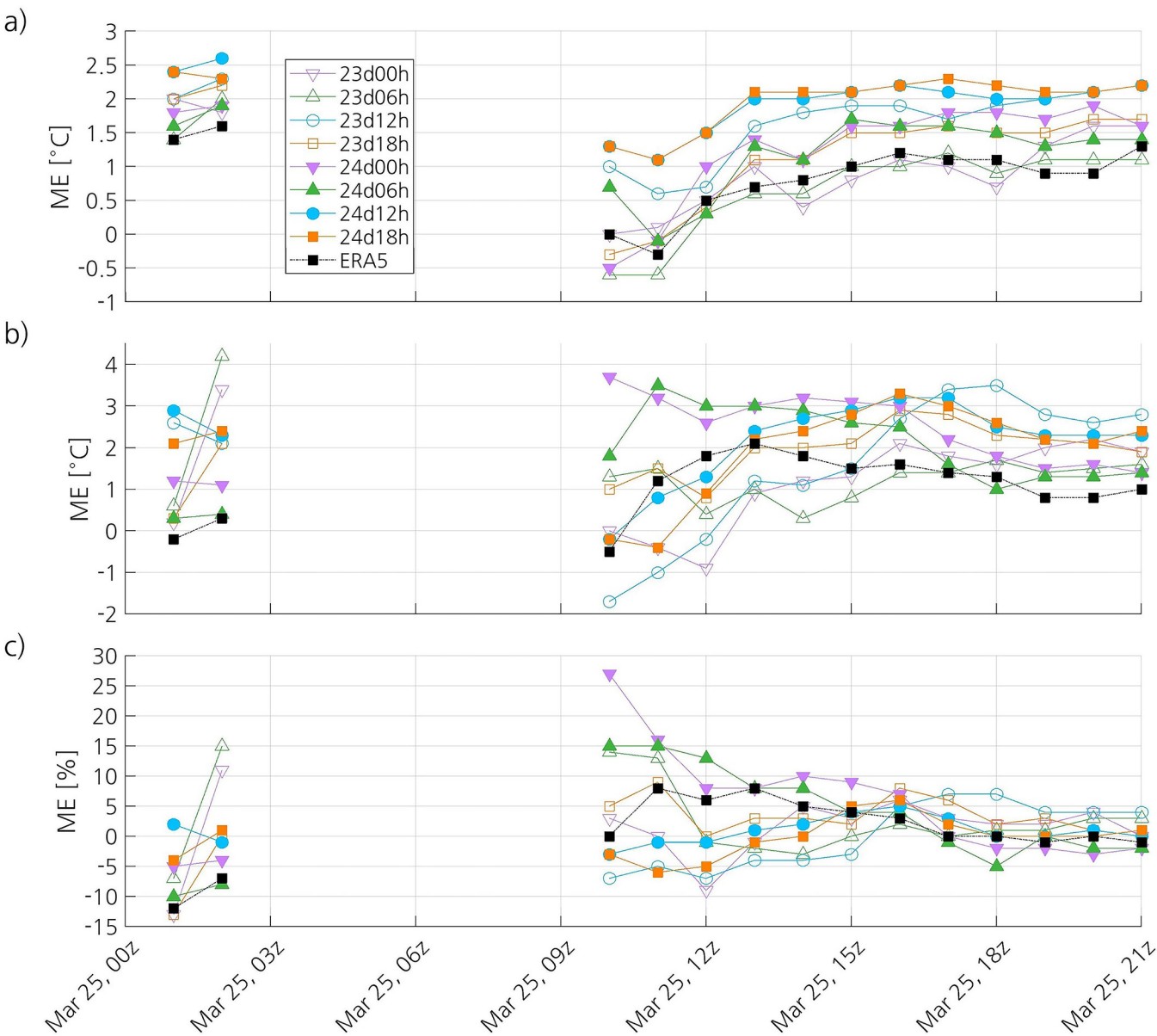

**Fig 10.** Time series of the ME of the simulated and ERA5 (a) 2-m temperature, (b) 2-m dewpoint temperature and (c) 2-m relative humidity. The datasets used as reference to compute the ME are the observations at surface stations.

(Fig 10B). In general, the RMSE of the simulations significantly increases from around 1000 UTC until some time between 1400 UTC and 1800 UTC, and then decreases (not shown). The Spearman correlation coefficient varies widely between simulations from 1000 UTC until 1300 UTC, and then it converges to values mostly between 0.4 and 0.6 (not shown).

ERA5 shows better skill at representing the dew point temperature than the simulations. In effect, the reanalysis has the lowest mean error (1.1˚C), the lowest RMSE (2.1˚C), and the highest Spearman correlation coefficient (0.73), indicating quite strong correlation with the observations (Table 4). The ME (Fig 10B) and the RMSE (not shown) of ERA5 decrease from around 1300 UTC until the PL has dissipated. During the last seven hours of the lifetime of the

**Table 4. Aggregate statistics computed to verify the simulated and ERA5 2-m dew point temperature against the observations from surface stations.**

|        | ME [°C] | RMSE [°C] | r    |
|--------|---------|-----------|------|
| 23d00h | 1.2     | 2.9       | 0.58 |
| 23d06h | 1.4     | 3         | 0.53 |
| 23d12h | 1.7     | 3.1       | 0.55 |
| 23d18h | 1.9     | 3.1       | 0.47 |
| 24d00h | 2.3     | 3.4       | 0.45 |
| 24d06h | 1.9     | 3.3       | 0.44 |
| 24d12h | 2.2     | 3.1       | 0.53 |
| 24d18h | 2       | 3         | 0.56 |
| ERA5   | 1.1     | 2.1       | 0.73 |

PL, the accuracy of ERA5 and the correlation coefficient are notably higher than that of the simulations (not shown).

**3.3.4 Relative humidity at 2 m.** The simulations that have captured the PL show somewhat better skill at representing the relative humidity than those that have not captured it (Table 5). The simulations 23d12h and 24d18h have the lowest mean error (0%), and 24d12h and 24d18h have the lowest RMSE (13%). The simulation 24d18h shows a weak correlation with the observations (0.41), and the other simulations have virtually no correlation with the observations. The time series of the ME (Fig 10C), the RMSE (not shown) and the Spearman correlation coefficient (not shown) differ between simulations, although 24d12h and 24d18h show a quite similar pattern. The ME of the simulations tends to converge with time, and the difference between them remains relatively small from around 1600 UTC on. The time series of the Spearman correlation coefficient shows correlation coefficients ranging from -1 to 1, and even the same simulation shows a wide range of correlation coefficients.

Compared to the simulations, ERA5 has higher accuracy (RMSE of 11%) and notably higher correlation with the observations (r of 0.62) (Table 5). However, its ME (1%) is only lower than that of half of the simulations. ERA5 has lower RMSE and higher correlation coefficient than the simulations from 1400 UTC until the PL has dissipated (not shown).

**3.3.5 Wind at 10 m.** The simulations that have not captured the PL show few values of wind speed over 15 m s$^{-1}$, whereas 24d12h and 24d18h show several values larger than 15 m s$^{-1}$ (not shown). In general, all simulations show some large overestimations and underestimations of wind speed, but the observed wind speeds over 20 m s$^{-1}$ are better captured by 24d12h and 24d18h. Most of the wind directions of the simulations that did not capture the PL are

**Table 5. Aggregate statistics computed to verify the simulated and ERA5 2-m relative humidity against the observations from surface stations.**

|        | ME [%] | RMSE [%] | r    |
|--------|--------|----------|------|
| 23d00h | 1      | 16       | 0.2  |
| 23d06h | 3      | 17       | 0.23 |
| 23d12h | 0      | 14       | 0.27 |
| 23d18h | 2      | 16       | 0.16 |
| 24d00h | 5      | 19       | 0.03 |
| 24d06h | 3      | 17       | 0.12 |
| 24d12h | 1      | 13       | 0.31 |
| 24d18h | 0      | 13       | 0.41 |
| ERA5   | 1      | 11       | 0.62 |

**Table 6. Aggregate RMSE-WVD computed to verify the simulated and ERA5 10-m wind against the observations from surface stations.**

|  | RMSE-WVD [m s$^{-1}$] |
|---|---|
| 23d00h | 7.1 |
| 23d06h | 7.8 |
| 23d12h | 6.2 |
| 23d18h | 6.3 |
| 24d00h | 6.6 |
| 24d06h | 6.8 |
| 24d12h | 5.9 |
| 24d18h | 4.9 |
| ERA5 | 3.7 |

located in the west-north-west/north-west/north-north-west (WNW/NW/NNW) quadrants of the wind rose (not shown), which correspond to the direction of the wind responsible for the MCAO. The main direction of the wind in 24d12h and 24d18h is NW, but these simulations also show winds coming from a wide range of directions, like the observed winds. However, the number of observations of the wind direction in the NW quadrant is much less compared to the simulations and ERA5, which is likely due to the fact that many wind observations are not represented in the wind rose because the recorded wind speed is zero. The simulation with the lowest RMSE-VWD is 24d18h (4.9 m s$^{-1}$), followed by 24d12h (5.9 m s$^{-1}$) and 23d12h (6.2 m s$^{-1}$) (Table 6). Overall, the RMSE-VWD of 24d12h and 24d18h increases from 1100 UTC to 1700 UTC, as the PL deepens (Fig 11A). The RMSE-VWD of 24d18h and 24d12h is lower than that of the other simulations during, respectively, almost all of the time and half of the time.

The wind rose of ERA5 is quite similar to that of 24d12h and 24d18h, the main difference being that ERA5 represents more frequent winds from the north-north-east (NNE) (not shown). ERA5 has the lowest RMSE-VWD (3.7 m s$^{-1}$), even when compared with the simulations that have captured the PL (Table 6). The time series of the RMSE-VWD of ERA5 is similar to that of 24d12h and 24d18h, with the difference that the RMSE-VWD of ERA5 is lower throughout the whole period (Fig 11A). At the end of the lifetime of the PL, the RMSE-VWD of all the simulations and ERA5 tends to converge to around 5 m s$^{-1}$.

Given that the simulation and ERA5 capture the observed SLP quite well, it is surprising that the skill of both at capturing the near-surface wind is not as good. This is likely due not only to model error, but also to representativeness error and observational error. In complex terrain, wind observations from weather stations may not be representative of the average wind over a larger area. In addition, measurements by anemometers are affected by topography, surface cover and surrounding obstacles [49]. The differences between the observed and simulated winds may be also due to the different period used to obtain the average wind in the different datasets.

**3.3.6 Maximum wind gusts at 10 m.** The ME of 24d12h and 24d18h is 1 m s$^{-1}$, and that of 23d12h is 0.1 m s$^{-1}$, whereas the ME of the simulations that have not captured the PL is negative or equal to zero (Table 7). The simulation 24d18h shows the lowest RMSE (4 m s$^{-1}$), followed by 24d12h (4.9 m s$^{-1}$). The other simulations have lower accuracy, with their RMSE ranging from 5.1 to 7.5 m s$^{-1}$. The simulations 24d12h and 24d18h have a quite strong correlation and strong correlation, respectively, with the observations, whereas all the others except for 24d00h have modest or virtually no correlation with the observations. The ME of the simulations that have captured the PL remains positive most of the time, whereas the ME of most of

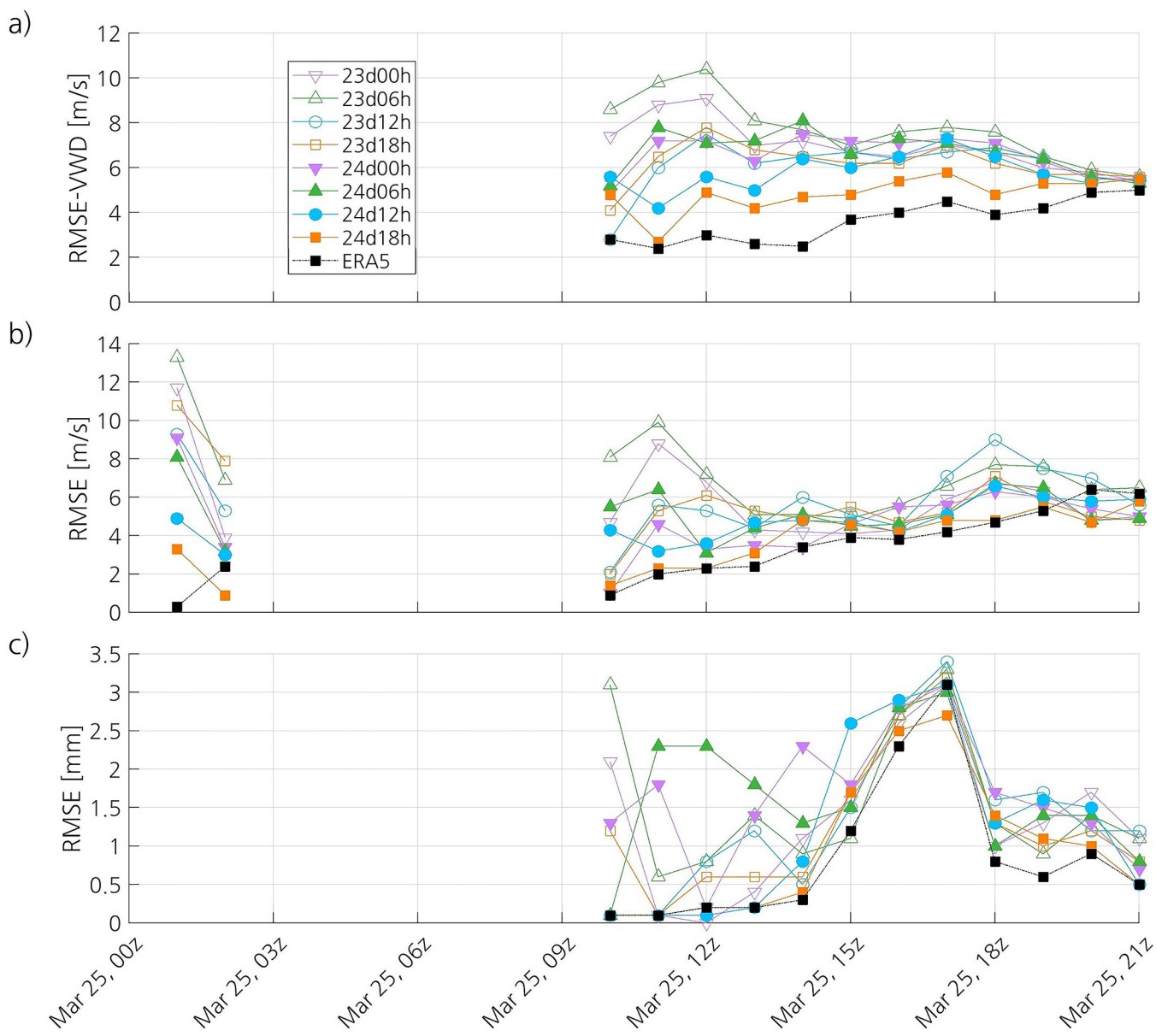

**Fig 11.** Time series of the (a) RMSE-VWD of the simulated and ERA5 10-m wind, and RMSE of the simulated and ERA5 (b) wind gusts and (c) 1-h accumulated precipitation. The datasets used as reference to compute the RMSE-WVD and RMSE are the observations at surface stations.

the other simulations is negative from 1600 UTC on (not shown). Overall, the RMSE of 24d12h and 24d18h increases from 1100 UTC on, and the latter has smaller RMSE than the rest of the simulations most of the time (Fig 11B). The time series of the Spearman correlation coefficient varies widely between simulations, with correlation coefficients ranging from -0.19 to 0.85, and even the same simulation shows a wide range of correlation coefficients (not shown).

The aggregate ME of ERA5 (1.9 m s$^{-1}$) is almost twice as that of the simulations that have captured the PL, but its RMSE (3.9 m s$^{-1}$) is lower and its correlation coefficient is higher (0.81) (Table 7). However, the difference in the RMSE and the correlation coefficient between ERA5 and 24d18h is very small. Like the simulations that have captured the PL, the ME of

**Table 7. Aggregate statistics computed to verify the simulated and ERA5 wind gusts against the observations from surface stations.**

|        | ME [m s$^{-1}$] | RMSE [m s$^{-1}$] | r |
|--------|-----------------|-------------------|------|
| 23d00h | -0.8 | 6.2 | 0.51 |
| 23d06h | -1   | 7.5 | 0.37 |
| 23d12h | 0.1  | 6.3 | 0.38 |
| 23d18h | -1.2 | 6.1 | 0.53 |
| 24d00h | -0.1 | 5.1 | 0.65 |
| 24d06h | 0    | 5.4 | 0.52 |
| 24d12h | 1    | 4.9 | 0.74 |
| 24d18h | 1    | 4   | 0.8  |
| ERA5   | 1.9  | 3.9 | 0.81 |

ERA5 remains positive most of the time (not shown). The time series of the RMSE of ERA5 follows closely that of 24d18h (Fig 11B). The correlation coefficient of ERA5 shows less variability than that of the simulations, remaining between 0.58 and 0.84 (not shown).

**3.3.7 1-h accumulated precipitation.** The aggregate statistics indicate that 24d18h has the most skill at representing precipitation (Table 8). The ME of the simulations is positive, with 23d12h and 24d18h having the lowest mean bias (0.1 mm). The latter also has the lowest RMSE (1.3 mm) and the highest correlation coefficient (0.53), indicating modest correlation with the observations. The other simulations have virtually no correlation with the observations, except for 24d12h, which shows weak correlation with them. During most of the time, the ME of all the simulations ranges from -1 to 1 mm (not shown). The highest values of RMSE are found at 16:00 and 17:00 UTC, just after the observed SLP minimum is attained (Fig 11C). There is a large spread of the correlation coefficients of the simulations (not shown).

In contrast with the simulations, ERA5 has a negative ME (-0.1 mm) (Table 8). The RMSE of ERA5 (1.2 mm) is lower than that of all the simulations, although it is only slightly lower than that of 24d18h. ERA5 shows modest correlation with the observations (0.62). Most of the time, ERA5 shows a somewhat smaller RMSE compared to the simulations (Fig 11C), and its correlation coefficients are higher than those of most of the simulations (not shown). The ERA5 1-h accumulated precipitation never exceeds 2.5 mm, which is likely due to its relatively low resolution. In contrast, the maximum simulated precipitation is 9.5 mm, corresponding to 24d12h, and the maximum observed precipitation is 20.6 mm. This agrees with the finding of Hu and Franzke [50] that ERA5 underestimates the daily precipitation extremes observed by weather stations in Germany.

**Table 8. Aggregate statistics computed to verify the simulated and ERA5 1-h accumulated precipitation against the observations from surface stations.**

|        | ME [mm] | RMSE [mm] | r |
|--------|---------|-----------|------|
| 23d00h | 0.2  | 1.6 | 0.22 |
| 23d06h | 0.3  | 1.8 | 0.22 |
| 23d12h | 0.1  | 1.6 | 0.35 |
| 23d18h | 0.2  | 1.5 | 0.34 |
| 24d00h | 0.5  | 1.8 | 0.19 |
| 24d06h | 0.5  | 1.8 | 0.23 |
| 24d12h | 0.2  | 1.6 | 0.42 |
| 24d18h | 0.1  | 1.3 | 0.53 |
| ERA5   | -0.1 | 1.2 | 0.62 |

## 3.4 Final discussion

To understand why only 24d12h and 24d18h correctly capture the development of the PL, it is necessary to analyse the simulated atmospheric fields from a few hours before its formation until its genesis time. It is assumed that, during this period, the synoptic conditions are favourable for PL formation in 24d12h and 24d18h, but not in the other simulations.

Fig 12 shows the SLP, the geopotential height at 500 hPa and the 1000-500 hPa thickness on 25 March at 0000 UTC. There is a 500-hPa through with a northeast-southwest orientation in all simulations. Although its shape is slightly different in the simulations, the through is in the

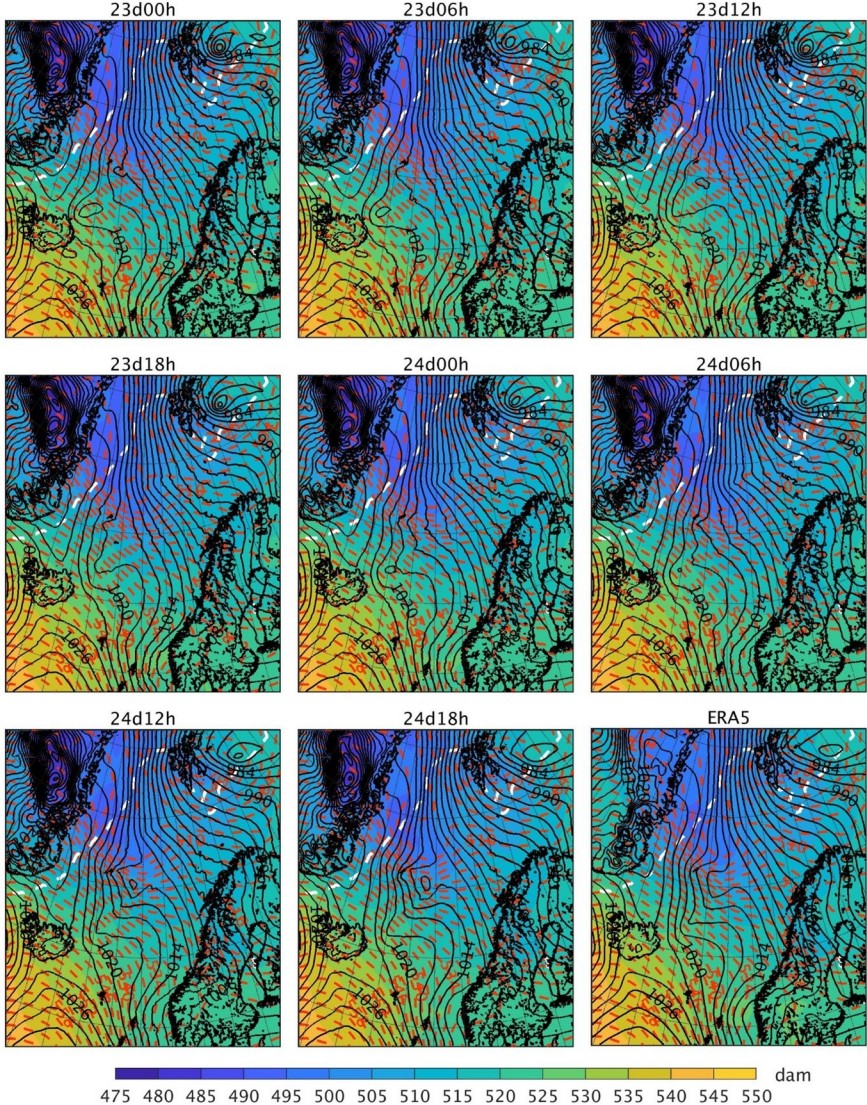

**Fig 12. Simulated and ERA5 fields showing the PL on 25 March 2019 at 0000 UTC.** The colourmap represents the 1000-500 hPa thickness (dam), the black isobars represent the SLP (hPa, contours every 2 hPa) and the red dashed lines represent the geopotential height at 500 hPa (dam, contours every 2 dam). The black outlining represents the coastline, and the white dashed line represents the sea ice edge, which is defined as the 0.15 contour of the sea ice concentration. ERA5 fields have been interpolated to the grid of the simulation using bicubic interpolation for the SLP, 1000-500 hPa thickness, and geopotential height, and bilinear interpolation for the sea ice concentration. The sea ice edge in ERA5 corresponds to the 25 March 2019 at 1200 UTC. The dataset used to plot the geographic contours has been obtained from the GSHHG [33] under a CC BY license, with permission from Dr. Paul Wessel.

same region and shows the same depth. The incipient PL in 24d12h and 24d18h with a well defined SLP minimum is located on the right side of this mid-tropospheric through, whereas in the other simulations only a weak (low-level) through within the SLP field in this area is observed. The 1000-500 hPa thickness field shows that the cold air tongue associated with the MCAO has a northwest-southeast orientation in all simulations.

Since the atmospheric conditions aloft are similar in all simulations during the genesis time of the PL, they cannot explain why it has only been correctly captured by 24d12h and 24d18h (with respect to both observations and reanalysis data). Therefore, the difference between the simulations must be in the lower atmosphere. Fig 13 shows the geopotential height, temperature and horizontal wind at 900 hPa on 24 March at 1900 UTC in the region where the low-level through preceding the genesis of the PL started to form (i.e. 5 hours before the PL shown in Fig 12). All simulations show a strong northwest-southeast temperature gradient to the west of Jan Mayen, close to the sea ice edge. In contrast with the other simulations, the northerly cold air advection and winds in 24d12h and 24d18h are more intense and more widely extended; therefore, the cold airmass moves further south in these two simulations. At the same time, on the east side of this cold air, a warm front pushes northward in these two simulations, with a more widely defined and stronger warm air advection in this area than the other simulations. These results indicate that, in the presence of a baroclinic environment, only the low-level atmospheric conditions with a well defined cold/warm air temperature advection present in 24d12h and 24d18h lead to baroclinic instability, which is involved in the genesis of the PL. It is also clear in Fig 13 that the low-level pressure deepens or vorticity (wind rotation) starts to develop in these two simulations, i.e. small scale features corresponding to the PL development phase.

Since 24d12h and 24d18h are the latest initialised simulations, their atmospheric fields during the hours preceding the PL formation are more similar to those of ERA5, the driving data, compared to those of the other simulations. Thus, the fact that the other simulations except for 23d12h do not represent the PL is due to forecast error growth and missing small-scale features during the initial stage of the PL formation. Nevertheless, the question remains about why 23d12h represents a PL at a later moment in time. In 23d12h, a strong low-level baroclinic zone forms a few hours before the PL forms in this simulation (Fig 14), and the PL shows baroclinic development. This PL makes landfall shortly after being formed, thus dissipating before it can reach a larger size. Fig 14 reveals also clearly that stronger winds over both cold (west) and warm (east) near the developed PL induce small scale conditions (i.e., temperature advections) favourable to strengthen low-level baroclinity and cyclogenesis in the latest initialised simulations, not present in the other simulations.

In conclusion, on the 24–25 March, the simulated environmental conditions are favourable for PL development, with a low-level baroclinic environment and an upper-level through, but the different evolution of the low-level circulation and small-scale features explains why a few simulations capture the PL whereas the others do not.

## 4. Conclusion

Compared to low-resolution models, convection-permitting models provide a better representation of physical processes [20]. Therefore, they are a powerful tool to study mesoscale phenomena, including PLs. This study has focused on a PL that made landfall in Norway in 2019, and the aim was to analyse the impact of the initial conditions on the simulation of the PL, and to analyse the skill of the CRCM6/GEM4 at reproducing it. The main limitations of this study is that the available conventional observations mostly cover the mature and dissipation stages of the PL, and that they are irregularly distributed in space.

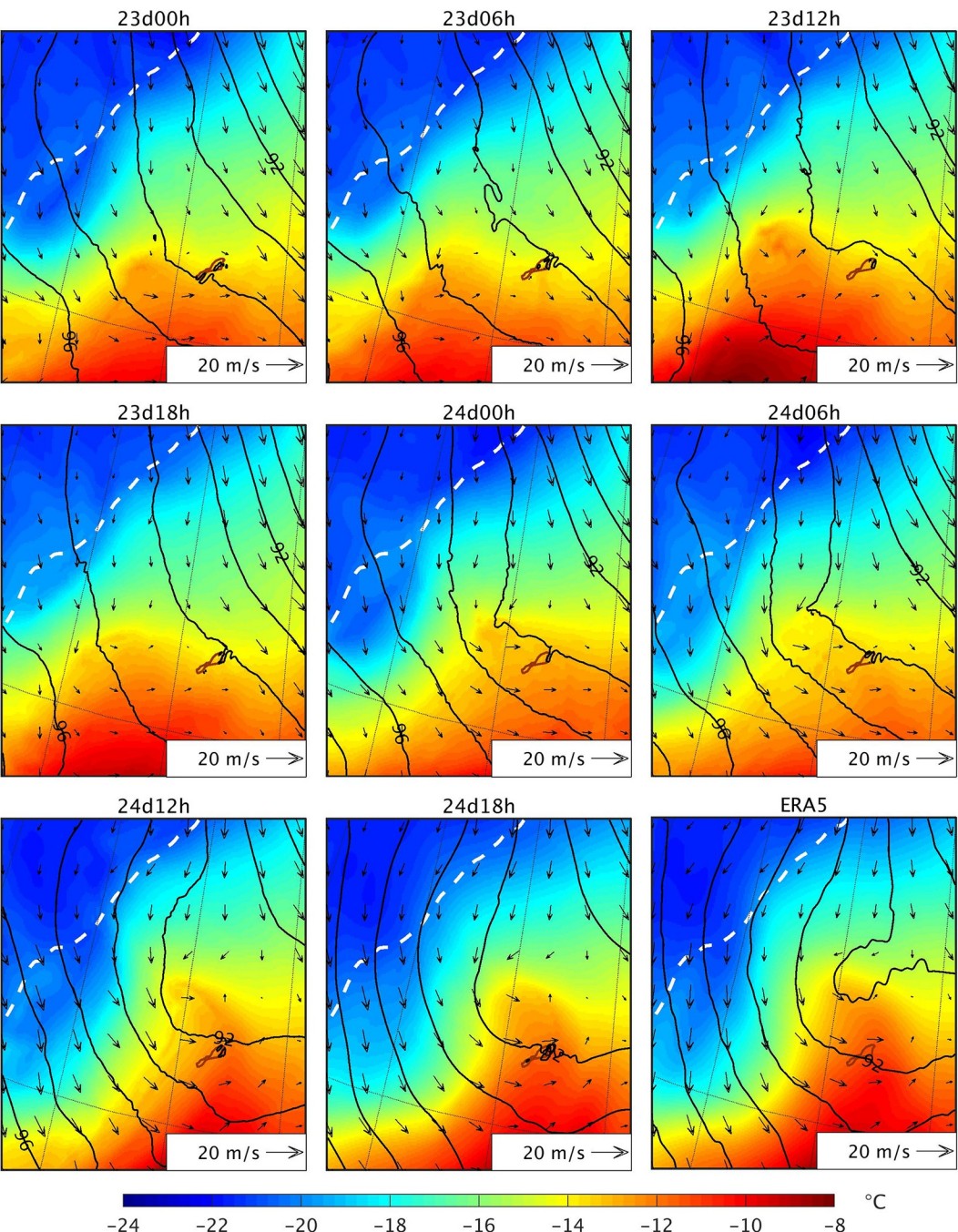

**Fig 13. Simulated and ERA5 fields on 24 March 2019 at 1900 UTC over the region around Jan Mayen.** The colourmap represents the temperature at 900 hPa (°C), the black isolines represent the geopotential height at 900 hPa (dam, contours every 1 dam), and the arrows represent the horizontal wind at 900 hPa. The white dashed line represents the sea ice edge, which is defined as the 0.15 contour of the sea ice concentration. ERA5 fields have been interpolated to the grid of the simulation using bicubic interpolation for the temperature and geopotential height, and bilinear interpolation for the horizontal wind and sea ice concentration. The sea ice edge in ERA5 corresponds to the 24 March 2019 at 1200 UTC. The dataset used to plot the geographic contours has been obtained from the GSHHG [33] under a CC BY license, with permission from Dr. Paul Wessel.

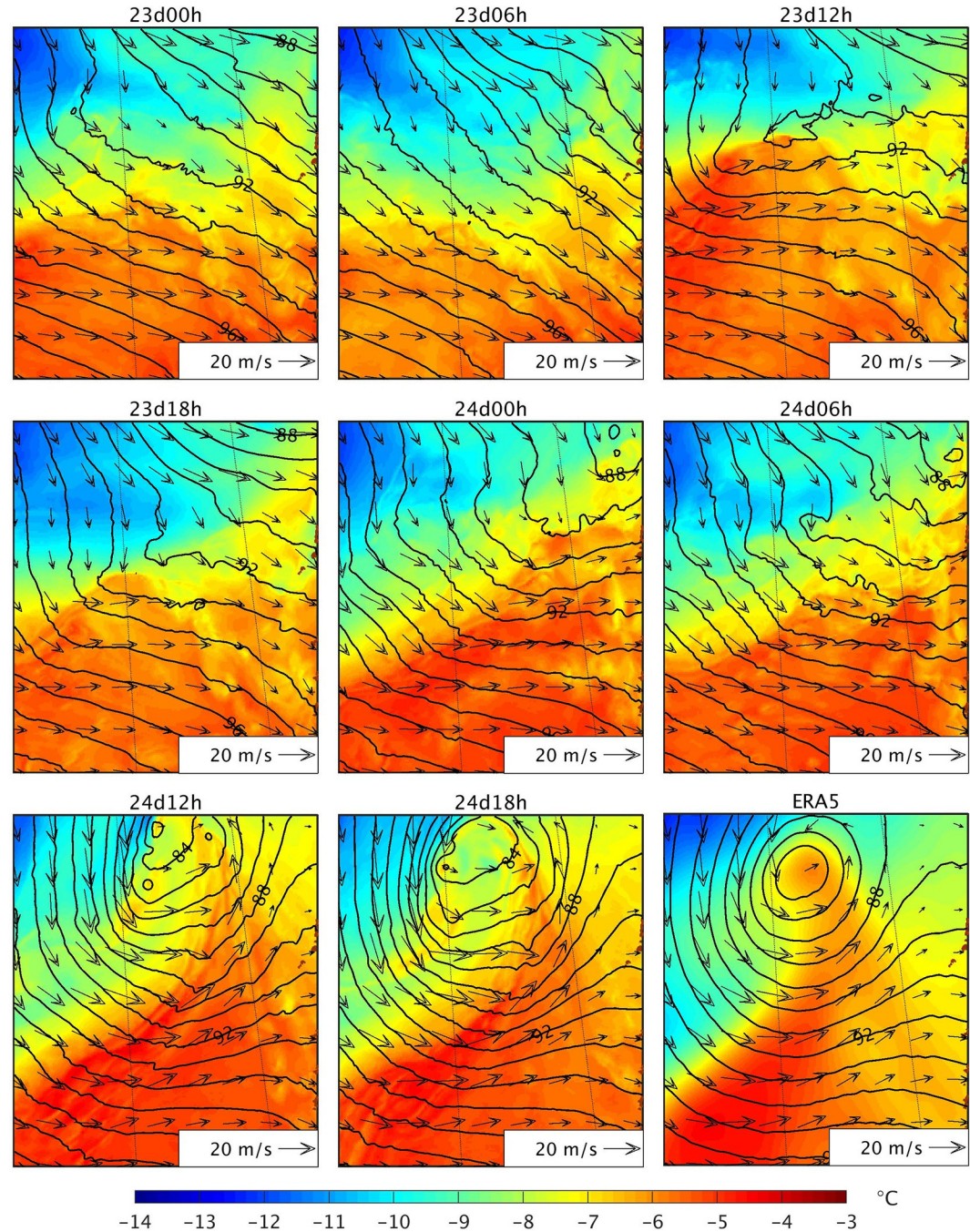

**Fig 14. Same as Fig 13, but for 25 March 2019 at 0900 UTC and over the region to the west of the Norwegian coast.**

One of the main findings of this study is that the ability of the CRCM6/GEM4 to capture the PL strongly depends on the initial conditions. In effect, only 23d12h and the latest initialised simulations 24d12h and 24d18h capture the development of the PL. The latter two represent well the lifetime, track and size of the observed PL. In contrast, the PL represented in 23d12h is much smaller than the observed PL, and its lifetime is less than half the lifetime of the observed PL. Further, the verification of the simulations against conventional observations

**Table 9. Added value of the CRCM6/GEM4 (simulation 24d18h) compared to ERA5 for the following variables: Sea level pressure (SLP), 2-m temperature ($T_{2m}$),** **2-m dew point temperature ($T_{d,2m}$), 10-m wind ($V_{10m}$), wind gusts (WG) and 1-h accumulated precipitation (PR).** The added value has been computed using the aggregate RMSE (RMSE-WVD for the 10-m wind), based on values presented in Tables 2–8. The added value computation is based on the study of Di Luca et al. [52; Equation 1)].

| SLP | $T_{2m}$ | $T_{d,2m}$ | RH | $V_{10m}$ | WG | PR |
|---|---|---|---|---|---|---|
| -0.8 hPa | -0.9˚C | -0.9˚C | -2% | -1.2 m s$^{-1}$ | -0.1 m s$^{-1}$ | -0.3 mm |

has shown that 24d18h has more skill than 24d12h at reproducing most of the near-surface variables analysed. These results indicate that the initialisation time has an important impact on whether the model captures or not this PL, and on how well it is represented. The two latest initialized simulations show northerly cold air advection and winds that are more intense than in the other simulations, leading to baroclinic instability and, subsequently, to the genesis of the PL. Nevertheless, since the environmental conditions– strong low-level temperature gradient and an upper-level through –on the 24 and 25 March are favourable for PL development, a PL can form at a later time if the low-level conditions are favourable for baroclinic instability to grow, which is what happens in 23d12h. In view of these results, it is suggested that future studies should investigate the potential to improve PL forecasts by using spectral nudging to maintain the low-level atmospheric fields and small scale features close to the driving data. Sensitivity tests should be conducted with different spectral nudging parameters and nudging horizontal wind, temperature, or both.

Another key finding is that the processes involved in the development of the PL need to be improved in the model in order to decrease the mean bias of the simulations that have captured it. Although all the statistics clearly show the better performance of 24d12h and 24d18h at reproducing SLP compared to the other simulations, it is notable that, for the other variables, these two simulations show similar or higher aggregate absolute mean bias. In particular, the parameterization of the surface heat fluxes in the CRCM6/GEM4 needs to be improved. In effect, the fact that 24d12h and 24d18h represent a PL deeper than the observed one, and show higher temperature mean bias compared to the other simulations and ERA5, seems to indicate that the ocean surface fluxes may be too strong.

Finally, the results have shown that ERA5 has more skill than the simulations, including those that have captured the PL, at reproducing the observed PL during its mature and dissipation stages. Table 9 shows the added value of the CRCM6/GEM4 compared to ERA5 when considering the best simulation (24d18h). For all the near-surface variables analysed here, the model does not provide added value in terms of accuracy (based on the RMSE values shown in Tables 2–8). It is surprising that the CRMC6/GEM4, a high-resolution model, does not provide added value compared to ERA5, the coarser reanalysis that drives it. There are two main reasons that could explain the fact that ERA5 has better skill than CRMC6/GEM4. First, conventional observations are assimilated into ERA5. Second, the verification of high-resolution simulations using standard statistics has some limitations. For instance, when verifying the simulation of a PL using dropsonde observations, Stoll et al. [17] found that a fuzzy verification method showed that the regional model AROME-Arctic had higher skill at capturing extreme values at small scales than the global model ECMWF HRES, whereas standard verification statistics were similar for both models. Finally, note that for this work we used GEM4, but a new version with improved physics parameterizations, GEM5, was recently released [51]. Therefore, an interesting course of research would be to analyse if this new version of GEM provides added value compared to ERA5 and to the current CRCM6/GEM4 simulations.

## Acknowledgments

Computations were made on the supercomputer Beluga, managed by Calcul Québec and the Digital Research Alliance of Canada. The authors are deeply indebted to Katja Winger for her essential support in the use of the CRCM6/GEM4, as well as for downloading and preparing available ERA5 reanalyses. The authors would also like to thank François Roberge for his valuable help in the use of the *r.diag* toolkit and the CRCM6/GEM4.

## Author Contributions

**Conceptualization:** Marta Moreno-Ibáñez, René Laprise.

**Formal analysis:** Marta Moreno-Ibáñez, René Laprise, Philippe Gachon.

**Funding acquisition:** René Laprise.

**Investigation:** Marta Moreno-Ibáñez.

**Methodology:** Marta Moreno-Ibáñez, René Laprise.

**Project administration:** René Laprise.

**Resources:** René Laprise.

**Software:** Marta Moreno-Ibáñez.

**Supervision:** René Laprise, Philippe Gachon.

**Validation:** René Laprise, Philippe Gachon.

**Visualization:** Marta Moreno-Ibáñez.

**Writing – original draft:** Marta Moreno-Ibáñez.

**Writing – review & editing:** Marta Moreno-Ibáñez, René Laprise, Philippe Gachon.

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
