## [Decision Letter · Decision Letter 0]

3 Jan 2023

PONE-D-22-29970Assessment of simulations of a polar low with the Canadian Regional Climate ModelPLOS ONE

Dear Dr. Moreno Ibáñez,

Thank you for submitting your manuscript to PLOS ONE. After careful consideration, we feel that it has merit but does not fully meet PLOS ONE’s publication criteria as it currently stands. Therefore, we invite you to submit a revised version of the manuscript that addresses the points raised during the review process. Please submit your revised manuscript by Feb 17 2023 11:59PM. If you will need more time than this to complete your revisions, please reply to this message or contact the journal office at plosone@plos.org. Please include the following items when submitting your revised manuscript:A rebuttal letter that responds to each point raised by the academic editor and reviewer(s). You should upload this letter as a separate file labeled 'Response to Reviewers'.A marked-up copy of your manuscript that highlights changes made to the original version. You should upload this as a separate file labeled 'Revised Manuscript with Track Changes'.An unmarked version of your revised paper without tracked changes. You should upload this as a separate file labeled 'Manuscript'.

We look forward to receiving your revised manuscript.

Kind regards,

Delei Li, Ph.D.

Academic Editor

PLOS ONE

Journal Requirements:

"This work was supported by the Discovery Grant program of the Natural Sciences and Engineering Research Council of Canada (NSERC) under Grant 707337, by the project “Marine Environmental Observation, Prediction and Response” (MEOPAR; http://meopar.ca) of the Networks of Centres of Excellence (NCE; http://www.nce-rce.gc.ca) of Canada, by the UQAM’s Faculty of Sciences under the programme “faculty financial support”, and by the excellence scholarship of the Trottier Family Foundation."

4. We note that Figures 1, 2, 3, 4 and 5 in your submission contain map/satellite images which may be copyrighted. All PLOS content is published under the Creative Commons Attribution License (CC BY 4.0), which means that the manuscript, images, and Supporting Information files will be freely available online, and any third party is permitted to access, download, copy, distribute, and use these materials in any way, even commercially, with proper attribution. For these reasons, we cannot publish previously copyrighted maps or satellite images created using proprietary data, such as Google software (Google Maps, Street View, and Earth). For more information, see our copyright guidelines: http://journals.plos.org/plosone/s/licenses-and-copyright.

a. You may seek permission from the original copyright holder of Figures 1, 2, 3, 4 and 5 to publish the content specifically under the CC BY 4.0 license.  

Reviewers' comments:

Reviewer's Responses to Questions

**Comments to the Author**

1. Is the manuscript technically sound, and do the data support the conclusions?

Reviewer #1: Yes

2. Has the statistical analysis been performed appropriately and rigorously? 

Reviewer #1: Yes

3. Have the authors made all data underlying the findings in their manuscript fully available?

Reviewer #1: Yes

4. Is the manuscript presented in an intelligible fashion and written in standard English?

Reviewer #1: Yes

5. Review Comments to the Author

Reviewer #1: Polar lows (PL) are always associated with severe weather conditions and its forecasting remains a challenge. This study focus on the impact of the initial conditions of the simulation of PL using the convection-permitting model of CRCM6/GEM4. This study shows some new phenomena. However, the physical mechanism behind the phenomenon is not explained here. Thus, the followings need to be commented or addressed before it is publishable.

General comments

1. This article is mainly focus on the description of phenomena, but without the exploration of physical mechanism. The author should give a discussion about the possible reason leading to the difference among the simulations.

2. ERA5 has more skill than the CPM simulation in producing the PL. This conclusion maybe related with the choice of parameterization schemes. Whether the authors have compared the simulations among the different parameterization schemes.

3. They compared the simulation with their driving forcing of ERA5 and found that the CPM does not seem to provide much added value compared to ERA5. However, they didn’t calculate the added value. I recommend that the authors should evaluate the simulation improvements with downscaling using the method proposed by Di Luca et al. (2013).

Di Luca A, Elía R, Laprise R. 2013. Potential for small scale added value of RCMs downscaled climate change signal. Clim. Dyn. 40: 601–618.

specific comments

1. Some of the sentence in this study may be result in confusion. For example, Line 26-29.

2. To much lines in figure 4, and it is difficult to identify.

6. PLOS authors have the option to publish the peer review history of their article (what does this mean?). If published, this will include your full peer review and any attached files.

Reviewer #1: No

---

## [Author Response · Author response to Decision Letter 0]

11 Aug 2023

Please see the attached PDF named "Response to Reviewers".

---

## [Decision Letter · Decision Letter 1]

14 Sep 2023

PONE-D-22-29970R1Assessment of simulations of a polar low with the Canadian Regional Climate ModelPLOS ONE

Dear Dr. Moreno Ibáñez,

Thank you for submitting your manuscript to PLOS ONE. After careful consideration, we feel that it has merit but does not fully meet PLOS ONE’s publication criteria as it currently stands. Therefore, we invite you to submit a revised version of the manuscript that addresses the points raised during the review process.

We look forward to receiving your revised manuscript.

Kind regards,

Delei Li, Ph.D.

Academic Editor

PLOS ONE

Journal Requirements:

Reviewers' comments:

Reviewer's Responses to Questions

**Comments to the Author**

1. If the authors have adequately addressed your comments raised in a previous round of review and you feel that this manuscript is now acceptable for publication, you may indicate that here to bypass the “Comments to the Author” section, enter your conflict of interest statement in the “Confidential to Editor” section, and submit your "Accept" recommendation.

Reviewer #1: All comments have been addressed

2. Is the manuscript technically sound, and do the data support the conclusions?

Reviewer #1: Yes

3. Has the statistical analysis been performed appropriately and rigorously? 

Reviewer #1: Yes

4. Have the authors made all data underlying the findings in their manuscript fully available?

Reviewer #1: Yes

5. Is the manuscript presented in an intelligible fashion and written in standard English?

Reviewer #1: Yes

6. Review Comments to the Author

Reviewer #1: The authors have revised the manuscript following the comments. And the manuscript has been improved. Thus, I'd like to recommend it to be accepted after delecting the sentences Line 11-15.

7. PLOS authors have the option to publish the peer review history of their article (what does this mean?). If published, this will include your full peer review and any attached files.

Reviewer #1: No

---

## [Author Response · Author response to Decision Letter 1]

14 Sep 2023

RESPONSE TO THE REVIEWER

Thank you very much for your valuable comments, which have helped improve the manuscript. Following your suggestion, we have deleted the sentences in Lines 11-15.

RESPONSE TO THE EDITOR

Following the suggestion of the reviewer, we have deleted the sentences in Line 11-15.

We have also reviewed our reference list to ensure that it is complete and correct. 

We have uploaded our figure files to the Preflight Analysis and Conversion Engine (PACE) digital diagnostic tool to ensure that our figures meet PLOS requirements.

---

## [Editor Report · Decision Letter 2]

18 Sep 2023

Assessment of simulations of a polar low with the Canadian Regional Climate Model

PONE-D-22-29970R2

Dear Dr. Moreno Ibáñez,

We’re pleased to inform you that your manuscript has been judged scientifically suitable for publication and will be formally accepted for publication once it meets all outstanding technical requirements.

Kind regards,

Delei Li, Ph.D.

Academic Editor

PLOS ONE
---

## [Editor Report · Acceptance letter]

25 Sep 2023

PONE-D-22-29970R2 

Assessment of simulations of a polar low with the Canadian Regional Climate Model 

Dear Dr. Moreno Ibáñez:

I'm pleased to inform you that your manuscript has been deemed suitable for publication in PLOS ONE. Congratulations! Your manuscript is now with our production department. 

Kind regards, 

on behalf of

Dr. Delei Li 

Academic Editor

PLOS ONE